# Non-volatile 2D MoS$_2$/black phosphorus heterojunction photodiodes in the near- to mid-infrared region

Yuyan Zhu [1,7], Yang Wang [1,2,3,4,7] ✉, Xingchen Pang[1], Yongbo Jiang[1], Xiaoxian Liu[1], Qing Li[2,5], Zhen Wang [2], Chunsen Liu [1,6], Weida Hu [2,5] ✉ & Peng Zhou [1,3,4,6] ✉

Cutting-edge mid-wavelength infrared (MWIR) sensing technologies leverage infrared photodetectors, memory units, and computing units to enhance machine vision. Real-time processing and decision-making challenges emerge with the increasing number of intelligent pixels. However, current operations are limited to in-sensor computing capabilities for near-infrared technology, and high-performance MWIR detectors for multi-state switching functions are lacking. Here, we demonstrate a non-volatile MoS$_2$/black phosphorus (BP) heterojunction MWIR photovoltaic detector featuring a semi-floating gate structure design, integrating near- to mid-infrared photodetection, memory and computing (PMC) functionalities. The PMC device exhibits the property of being able to store a stable responsivity, which varies linearly with the stored conductance state. Significantly, device weights (stable responsivity) can be programmed with power consumption as low as 1.8 fJ, and the blackbody peak responsivity can reach 1.68 A/W for the MWIR band. In the simulation of Faster Region with convolution neural network (CNN) based on the FLIR dataset, the PMC hardware responsivity weights can reach 89% mean Average Precision index of the feature extraction network software weights. This MWIR photo-voltaic detector, with its versatile functionalities, holds significant promise for applications in advanced infrared object detection and recognition systems.

In the era of artificial intelligence and the Internet of Things (IoT), mid-wavelength infrared (MWIR) object detection and identification technology gains significance in advanced assisted driving, event monitoring, complex environmental reconnaissance and other fields[1]. Traditional MWIR photodetectors, using materials like mercury cadmium telluride (HgCdTe) and indium antimonide (InSb), necessitate refrigeration for temperature control. This poses challenges for low-cost, portable instrumentation, leading to energy waste and bulkiness[2–5]. Consequently, conventional MWIR detection systems encounter challenges in power consumption, latency, and functional integration, both in terms of architecture and device structures.

Two-dimensional (2D) materials offer unique advantages, allowing diverse functional layers due to a rich material system. The layered surface lacks hanging bonds, and single atomic level thickness, along

[1]State Key Laboratory of ASIC and System, School of Microelectronics, Fudan University, Shanghai 200433, China. [2]State Key Laboratory of Infrared Physics, Shanghai Institute of Technical Physics, Chinese Academy of Sciences, Shanghai 200083, China. [3]Shaoxin Laboratory, Shaoxing 312000, China. [4]Shanghai Frontiers Science Research Base of Intelligent Optoelectronics and Perception, Institute of Optoelectronics, Fudan University, Shanghai 200433, China. [5]Hangzhou Institute for Advanced Study, University of Chinese Academy of Sciences, Hangzhou 310024, China. [6]State Key Laboratory of Integrated Chip and System, Frontier Institute of Chip and System, Fudan University, Shanghai 200433, China. [7]These authors contributed equally: Yuyan Zhu, Yang Wang. ✉e-mail: yang_wang@fudan.edu.cn; wdhu@mail.sitp.ac.cn; pengzhou@fudan.edu.cn

with van der Waals (vdWs) contact between layers, supports high-density heterogeneous integration[6–9]. Simultaneously, 2D materials exhibit high photon sensitivity, achieving an MWIR photodetector with elevated responsivity at room temperature[10,11]. In system architecture, efforts are being made to integrate flash memory with ultrafast programming speed and computing, enabling direct input of analog current signals from the image sensor to a crossbar array with stored conductance weights. This facilitates multiply and accumulate computing (MAC) used in image recognition neural networks, reducing power consumption and latency in memory access[12–14]. Challenges persist in the energy consumption related to analog signal conversion and transmission between the photodetector and in-memory computing[15,16]. To address this, researchers explore combining sensors with in-memory computing structures. The all-in-one hardware combines sensors and memory calculations for motion detection, with a calculation process that deviates from regular MAC operations.[17] Certain hardware serves both as an image sensor and an in-memory computing unit, utilizing Ohm's law for processing image sensing data[18]. The sensor array transforms the optical information (incident illumination power (P)) of the image into photocurrent. However, the current signal requires conversion to a voltage signal for input into the processing array (identical to the photosensitive array), leading to non-simultaneous sensing and processing. Another system structure integrates sensing and processing with responsivity serving as the weight. The incident illumination power of the image serves as the input, and the output is the MAC calculation result based on the $I = RP$ law[19,20], where $I$ represents the output photocurrent and $R$ represents the photoresponsivity. This system architecture is only suitable for circuit connections in fully connected networks. The existing issues necessitate an all-in-one integrated innovation paradigm integrating high-performance MWIR detectors, weight storage, and advanced computing is desired but not yet realized.

Here, we leverage the heterogeneous integration properties of two-dimensional materials to effectively combine the characteristics of flash memory and MWIR photovoltaic detection. This integration empowers the fabrication of hardware designed for the MAC operation, facilitating efficient and low-power execution for MWIR target detection and recognition. A key capability of this device is the ability to store and modify responsivity, essentially representing the weights of neural networks. The memory structure is realized using ultra-fast $MoS_2/h$-BN/graphene flash memory[21], ensuring efficient and rapid storage operations. The photosensitivity from near- to mid-infrared at room temperature is achieved through the $BP/MoS_2$ heterojunction. Notably, the pulse width required for changing the weight is remarkably low, measuring only 20 ns. Additionally, the dynamic range of electrical storage exceeds $10^6$. Verification through MWIR blackbody radiation testing in an actual scene confirms the device's effectiveness. The peak responsivity is approximately 1.68 A/W at 3.6 μm and diminishes to half at 3.9 μm. A noteworthy outcome is the demonstrated effectiveness of the PMC device's responsivity weight in replacing the weight of the feature extraction neural network. By replacing 100% of the weights of the convolutional layers, the mAP of the device weights can reach 89% of the software training.

## Results

### MWIR PMC device based on CNN connection

Advanced Driving Assistance System (ADAS) involves the use of various millimeter-wave radar, LiDAR, single/binocular cameras and satellite navigation installed in the car to continuously sense the surrounding environment during driving. These photodetectors collect data for object detection and recognition, enhancing the driver's awareness of potential hazards in advance. In the ADAS environment, the ability to perceive mid-wave thermal infrared radiation provides unique complementary advantages to existing sensor technology. We aim to leverage the verified and analyzed near- to mid-infrared non-

volatile in-sensor computing devices to detect and distinguish pedestrians, cyclists and motor vehicles under extreme weather conditions such as complete darkness and bad weather[6]. In Fig. 1a, the PMC device embedded in the car captures the MWIR light emitted by the object at night. It stores the photoelectric information in a way similar to the memory of the human brain, enabling subsequent detection and recognition of the object to assist in the final decision-making process. To solve and optimize the problems of image processing and recognition, many researches have achieved remarkable success. As illustrated in Fig. 1b, the detector array transforms the optical information (light power) of the image into the photocurrent. However, the current signal must undergo a conversion into a voltage (V) signal for further processing. This conversion involves the analog-to-digital converter (ADC), which samples the current and converts it into a digital signal. Subsequently, the digital-to-analog converter (DAC) transforms the digital signal into the corresponding voltage signal[22]. And then the MAC operation can be conducted based on the $I = VG$ (G represents conductance) and Kirchhoff's Current Law. It is important to note that sensing and processing are not simultaneous in this configuration[18]. In Fig. 1c, a system structure in which sensing and processing are integrated with the responsivity as the weight. The light power of the image serves as the input, and the output is the calculation result of the MAC based on $I = RP$ ($I$ represents photocurrent, $R$ represents responsivity and $P$ represents light power) law[19]. This integration allows for simultaneous sensing and processing, eliminating the need for transmission costs between them. In this way, the sensing and processing are simultaneous and the transmission cost between them can be eliminated. However, the responsivity of the device in the array cannot be effectively stored and must be maintained by additional gate voltages.[19] Due to the limitations imposed by the circuit connection form of the array, this structure corresponds only to the fully connected neural network (FCN). Consequently, as the complexity of image recognition increases (with a rapid increase in the number of weights), there are high demands on the total number of devices and the integration degree. Therefore, a circuit connection suited for CNN is required. The circuit connection of PMC devices standing for the kernel of the CNN is illustrated in Fig. 1d. The equivalent circuit for this connection consists of devices in parallel between the positive power supply and the ground. Each device can function as an input port for a power density. The total current flowing between the positive power source and the ground is the sum of the product of the responsivity of each device and the corresponding light power density. To achieve this, the conductance states of the devices need to be altered by an electrical pulse. The infrared responsivity corresponding to the conductance state increases linearly (it's worth noting that the conductance state cannot be changed by infrared light, only by electrical pulses), depicted in Fig. 1e. This characteristic ensures that the responsivity weights of the device can be accurately programmed by adjusting the conductivity value. Furthermore, the net photocurrent is linearly related to the incident power density shown in Fig. 1f. In other word, the responsivity should be stable when the input light power density changes. This feature ensures that the weights can be stably mapped on the responsivity to accurately complete the MAC operation.

### Non-volatile infrared photovoltaic detectors

In Fig. 2a, the PMC device is presented as a layered heterojunction, comprising four 2D-material layers: black phosphorus (BP), molybdenum disulfide ($MoS_2$), $h$-BN (tunneling layer) and graphene (the reason of semi-floating gate can be found in Supplementary Fig. 10). The $BP/MoS_2$ heterojunction serves as the channel material. Detailed methods for device preparation and corresponding experimental procedure are provided in Supplementary Fig. 1 and Methods. This structure can be understood in two main parts: the left part features the $MoS_2/h$-BN/ graphene flash memory structure utilized to regulate

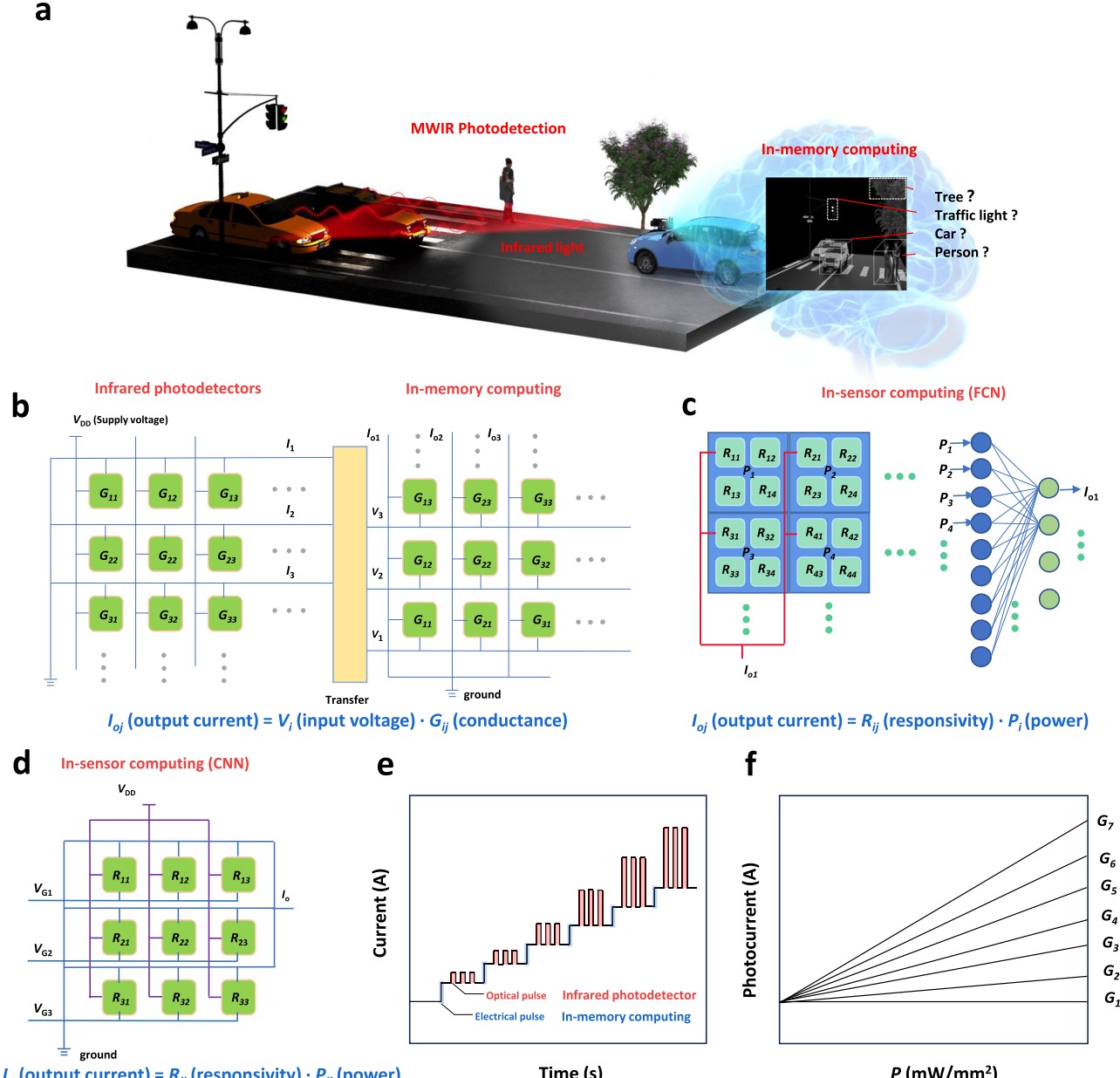

**Fig. 1 | MWIR PMC device based on CNN connection. a** PMC (photodetection, memory and computing) hardware assisted driving at night schematic diagram. MWIR represents mid-wave infrared. **b** Circuitry integrating infrared photo-detectors and in-memory computing functions in the same array. **c** In-sensor computing based on the fully connection neural network (FCN). In the subscript, *i* represents the row sequence, and *j* represents the column sequence. **d** Circuit connection of PMC devices standing for the kernel of the convolution neural network (CNN). $V_G$ represents the programming voltage on the gate electrode. **e, f** Ideal characteristics of the PMC device include the capability to alter conductance states through electrical pulses, with a corresponding linear increase in mid-wave infrared responsivity aligned with the conductance state (not influenced by incident light). Furthermore, the net photocurrent exhibits a linear relationship with the incident power density.

the conductance of the channel and maintain the conductance when the gate bias voltage ($V_{BG}$) is withdrawn. The right part, consisting of the BP/MoS$_2$ heterojunction, is dedicated to infrared photovoltaic detection. Attempts were made to use a single MoS$_2$ and BP material as the channel[23], but the infrared light caused changes in the conductance state with a limited modulated dynamic range (details in Supplementary Figs. 6 and 8). Consequently, we opted for the BP/MoS$_2$ hetero-junction as the channel material (the photoresponse of single BP/MoS$_2$ heterojunction can be seen in Supplementary Fig. 7). The choice of material thickness is crucial for optimal device performance (the effect of thickness can be seen in Supplementary Fig. 9). The thinner MoS$_2$ material is favorable for storing a large dynamic range, while the

thicker MoS$_2$ exhibits better infrared photoresponse[24], so reasonable control of the material thickness to achieve the ideal device perfor-mance is needed (Detailed thickness in Supplementary Fig. 2). Figure 2b illustrates the scanning transmission electron microscope (STEM) image corresponding to the device structure in Fig. 2a, show-casing clean and flat interfaces that contribute to good performance. Energy dispersive X-ray (EDX) spectroscopy mapping and Raman spectra are provided in Supplementary Figs. 3 and 4. As shown in Fig. 2c, the bi-directional scanning of $V_{BG}$ reveals a distinct clockwise memory window, with the channel current measured at $V_{DS} = 1$ V. The memory window achieves a range of 30 V when the $V_{BG}$ swept between ±35 V and the dynamic range[25] is more than 10$^6$, providing numerous

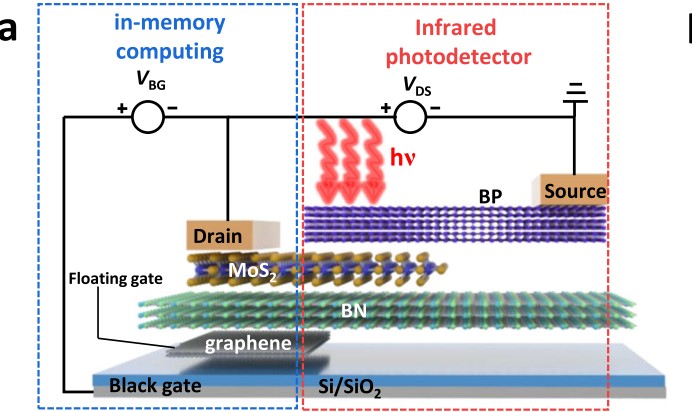

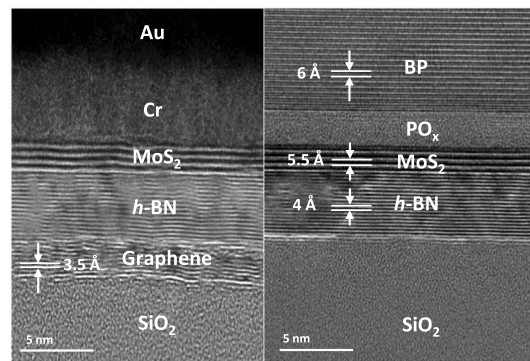

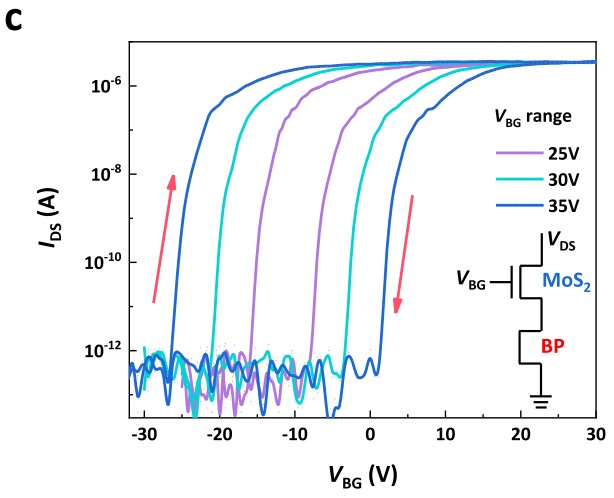

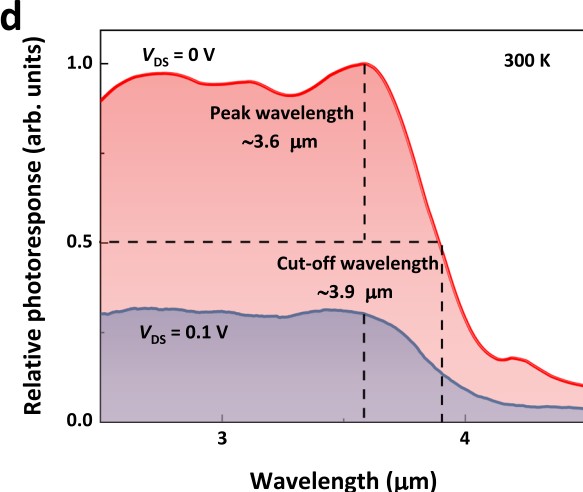

**Fig. 2 | Non-volatile MWIR in-sensor computing device. a** Device structure of BP (black phosphorus)/MoS₂/*h*-BN (boron nitride hexagonal)/graphene vdWs (van der waals) heterostructure. Memory structure (the left part) consisting of MoS₂ (channel material), *h*-BN (tunneling layer), graphene (floating gate layer) and photodetection structure (the right part) formed by BP/MoS₂ heterojunction co-constitute the non-volatile in-sensor computing device. **b** Scanning transmission electron microscope image corresponding to the device structure demonstrates the flat and clean interfaces. The scale bar is 5 nm. **c** Transfer characteristics of the device controlled by the back gate. The memory window caused by the shift of the threshold voltage reaches 30 V at $V_{DS}$ (drain-source voltage) = 1 V when $V_{BG}$ (backgate-source voltage) sweeps from −35 V to 35 V and owns a dynamic range of more than 10⁶. **c** The circuit model of the device channel. **d** Blackbody radiation response reaches a maximum of 3.6 µm. Source data are provided as a Source Data file (https://doi.org/10.6084/m9.figshare.25930456).

intermediate conductance states. The photocurrent spectrum of the device in the MWIR band was obtained by Fourier transform infrared spectroscopy (FTIR)[26]. For photodetection in real scenes, relative photo-response at various wavelengths is measured in Fig. 2d. The blackbody radiation response peaks at 3.6 µm, making it suitable for MWIR detection. And the relative photo-response is halved at 3.9 µm.

**Infrared laser response at various conductance states**

The retention of multi-conductance states for 1000 s tested at $V_{DS}$ = 1 V is demonstrated in Fig. 3a. Initially, a positive pulse (amplitude 30 V) is applied to the $V_{BG}$ input to ensure that the device is reset to the off-state with low conductance. Subsequently, it is programmed by a negative pulse (amplitude −30 V, width 20 ns). Besides, the fundamental memory characteristics can be seen in Supplementary Fig. 11. The voltage for nanoseconds duration is a high-frequency signal and the parasitic impedances could increase the effective duration of the voltage pulse. To address this concern, we utilized an oscilloscope to measure the actual ultrafast pulses received by the PMC device. The corresponding test results, illustrating the actual pulse waveform, are shown in Supplementary Fig. 12. Varying the number of the $V_{BG}$ pulses allows the device to be programmed to more than 12 stable and distinguishable states (Pulse

modulation under various pulse sets can be found in Supplementary Fig. 13) with a long retention time exceeding 10³ s. Additionally, we make a clarification on the methodology for determining the retention time and performed exponential fitting of Ids against time, which shows that the high and low resistance states of the device can be held for up to 10 years. Furthermore, the degradation of the device can be effectively curbed by coating the BP materials surface with a passivation layer of PMMA (Supplementary Figs. 15–17). Figure 3b illustrates the maximum range of conductance value change, confirming that the device functions well after 10⁴ cycles of periodic pulse signals at the $V_{BG}$ input (negative: amplitude −30 V, width 20 ns; positive: amplitude 30 V, width 50 ns). The programming involves using −30 V/20 ns to set the device to a high resistance state (HRS) and 30 V/50 ns to set it to a low resistance state (LRS)[27]. All the channel current is read out at $V_{BG}$ = 0 V, $V_{DS}$ = 1 V. After 10⁵ pulse cycles, the low conductivity state fails to 2 nA (details can be found in Supplementary Fig. 14b). Real-time tests for conductance configuration and the photo-response at different conductance states are illustrated in Fig. 3c. Two phenomena are verified: as the conductance increases, the net photocurrent ($I_{ph}$) also increases under the same laser power density of 16 µW/µm² (strong enough for saturation photocurrent) with the wavelength of 1550 nm;

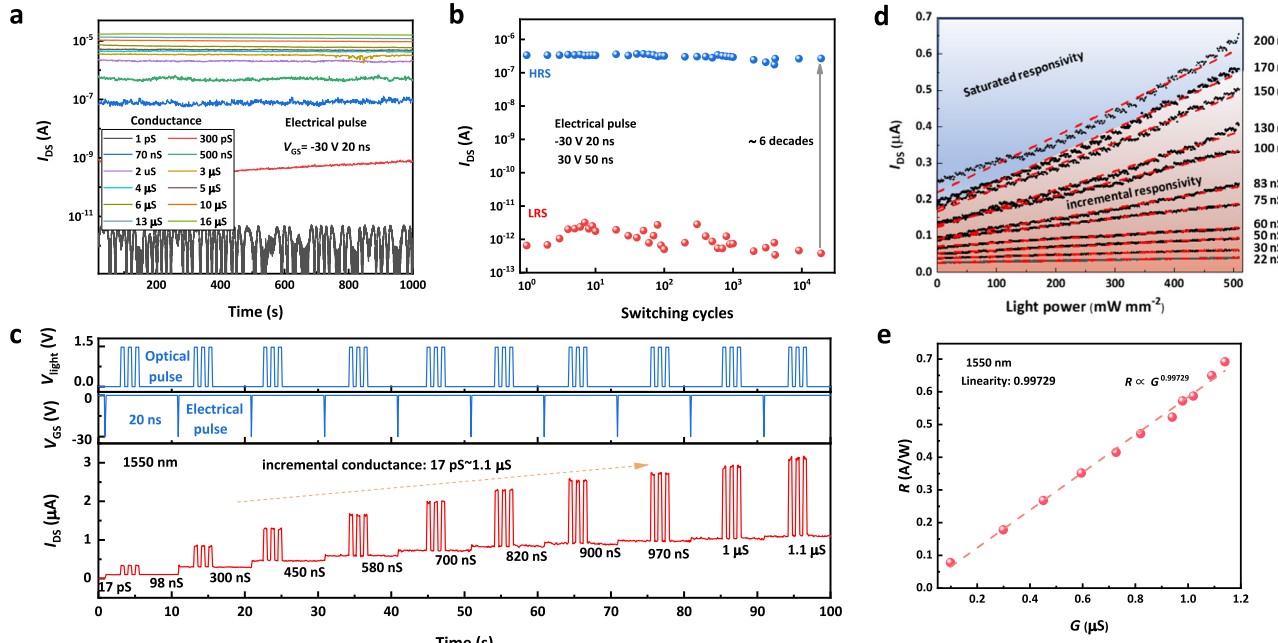

**Fig. 3 | Infrared laser response of the PMC device at various conductance states.** **a** Multi-conductance states (1 pS to 16 µS) retention for 1000 s measured at $V_{DS} = 1$ V. **b** Device endurance measurement. The −30 V/20 ns pulse is used to program the device to a high resistance state (HRS), and the 30 V/50 ns pulse is used to program the device to a low resistance state (LRS). The channel current is read out at $V_{BG} = 0$ V, $V_{DS} = 1$ V. **c** Real-time test for conductance configuration and the photo-response at different conductance states (17 pS to 1.1 µS). $V_{light}$ is the drive voltage of the laser. Its amplitude is linearly related to the output power of the laser. $V_{GS}$ is the gate-source voltage. The laser pulse set (wavelength: 1550 nm,

power density: 16 µW/µm² ($V_{light} = 1.5$ V), frequency: 1 Hz); The electrical pulse set (amplitude: −30 V, pulse width: 20 ns, frequency: 0.1 Hz). **d** Stable responsivity of different conductance states at 1550 nm laser. The dots are the actual measurements and the dashed line is the linear fit curve. The red and blue shaded areas represent the region of incremental and saturated responsivity. **e** Influence of conductance on the corresponding responsivity. The red dashed line shows the fitting result: the responsivity linearly and positively correlated with the conductance (linearity: $R^2 = 0.99729$). Source data are provided as a Source Data file (https://doi.org/10.6084/m9.figshare.25930456).

additionally, it has been observed that the conductance states remain unchanged when exposed to the infrared laser at 1550 nm. The reconfiguration procedure of the conductance state can be found in Supplementary Fig. 14a. Per-programming energy (PPE) can be calculated based on PPE = $V_{pulse} I_{gate} t_{pulse}$ = 1.8 fJ ($I_{gate}$ is the gate leakage current when programming: 3 nA). Similar measurements were conducted under the wavelength of 638 nm, 1064 nm and 1310 nm, as illustrated in Supplementary Figs. 18–20. In Fig. 3d, the change of $I_{ph}$ with power density at different conductance levels ranging from 22 to 200 nS is examined. The trend shows that the $I_{ph}$ increases approximately linearly with the power density. The slope of the linear relationship is the responsivity of the corresponding conductance state. The responsivity increases with increasing conductance state, eventually reaching saturation (the relationship between responsivity and power density can be found in Supplementary Fig. 22). Besides, similar measurement under 1310 nm can be found in Supplementary Fig. 23. Upon calculation ($R = I_{ph}/P$), the corresponding relationship between responsivity and conductance is obtained in Fig. 3e. Furthermore, the red dashed line represents the fitting result, indicating that the responsivity is linearly and positively correlated with the conductance (linearity: $R^2 = 0.99729$) (the fitting results of 1064 nm and 1310 nm can be found in Supplementary Fig. 21). These results described above mean that we can obtain various and stable responsivity which corresponds linearly to conductance, and the responsivity of the PMC device can be changed by applying ultrafast (20 ns) $V_{BG}$ pulses. For the mid-wave infrared range, the responsivity does not correspond linearly to the conductance states. This non-linearity increases the computational effort of in-situ training, thereby extending the time to train the neural network. An exponential fitting approximation can be used to establish a fixed function

mapping relationship, which will facilitate the mapping of weights and improve training efficiency (see Supplementary Fig. 26).

## Blackbody detection and mechanism analysis

Blackbody detection in various conductance states reflects irregular infrared radiation and weak signals from real objects, which is essential to reliably evaluate their potential in practical detection and recognition. The characterization diagram for blackbody detection is shown in Supplementary Fig. 27. The peak spectra radiance of the corresponding blackbody at a blackbody temperature of 1000 K lies in the MWIR band. The photocurrent and responsivity of the PMC device at various conductance states under 1000 K blackbody temperature indicate that the MWIR photovoltaic detector has good blackbody detection and polymorphic modulation capabilities (Fig. 4a and Supplementary Fig. 28). The blackbody responsivity ($R_{blackbody}$) of PMC devices increases from 0.125 to 0.512 A/W with increasing conductivity state (see Supplementary Section 6 for further details of the calculations). Fourier transform infrared spectroscopy (FTIR) was used to obtain the relationship between the responsivity of the PMC device and the radiation wavelength (Fig. 2d and Supplementary Fig. 29a). Further, we can obtain the responsivity ($R_{(λ)}$) and external quantum efficiency (QE) of the device at different wavelengths by changing the conductance state, indicating that higher conductance states correspond to higher responsivity (Fig. 4b, c and Supplementary Fig. 29c). The ($R_{(λ)}$) and QE of the PMC device are calculated as high as 1.68 A/W and 60% in 3600 nm under blackbody radiation at room temperature. The maximum specific detectivity of the tested device is close to $10^9$ cm Hz$^{1/2}$ W$^{-1}$ in the mid-infrared (3.5 µm) when the conductance state changes (see Supplementary Fig. 30). High-performance, non-volatile MWIR

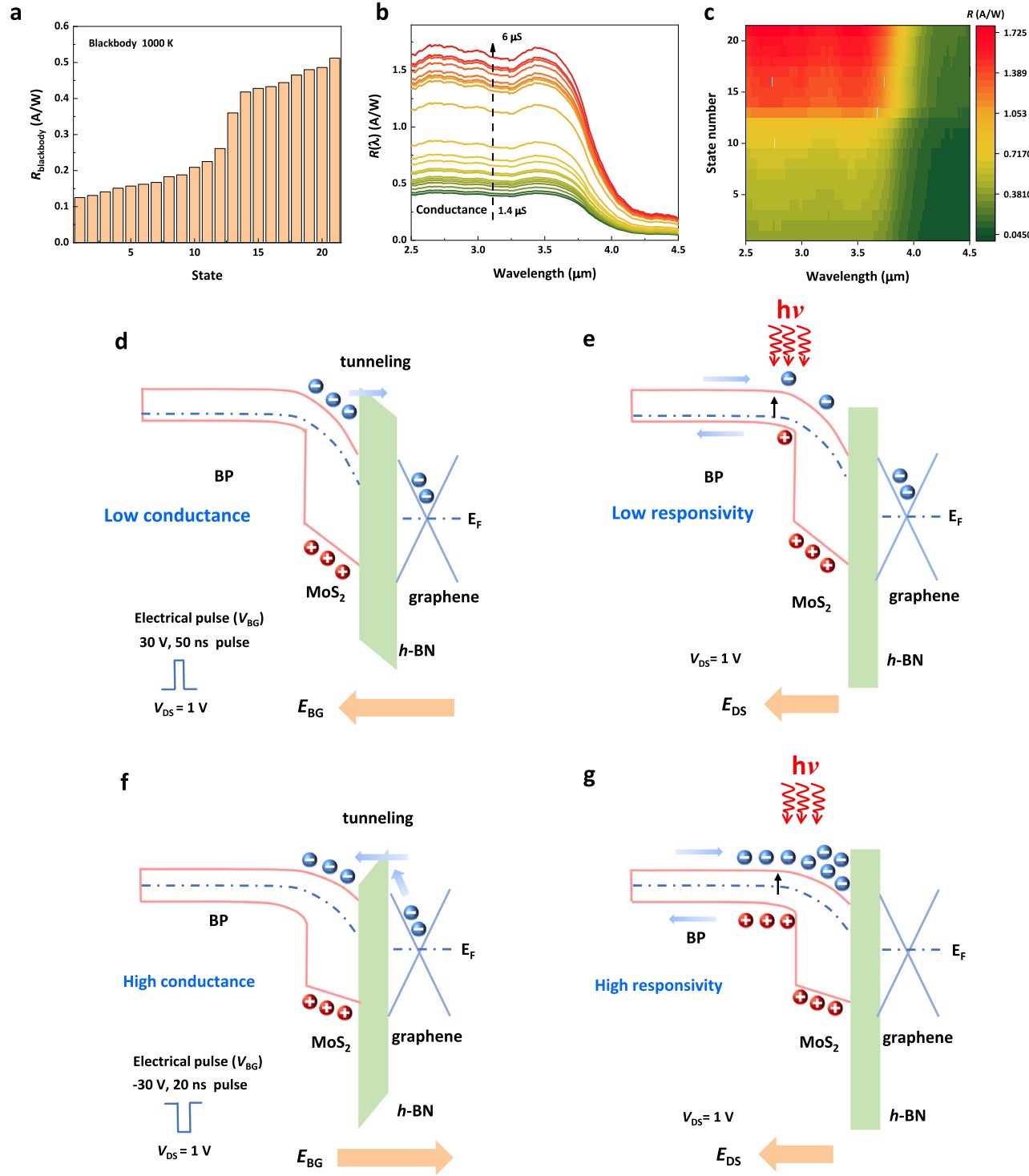

**Fig. 4 | Blackbody detection and mechanism analysis of the PMC device at various conductance states. a** Blackbody responsivity of PMC devices at various states. The temperature of the blackbody is 1000 K. The channel current is read out at $V_{DS}$ = 0.1 V. **b** Responsivity as a function of wavelength at various conductance states. Conductance state change from 1.4 to 6 μS at $V_{DS}$ = 0.1 V. The $R_{(\lambda)}$ of the PMC devices is calculated as high as 1.68 A/W in 3600 nm under blackbody radiation. **c** Responsivity mapping plots at different wavelengths for different conductivity states. **d** Formation of low conductivity states. **e** Lower infrared photoresponse in low conductance states. **f** Formation of high conductance states. **g** Higher infrared photoresponse in high conductance states. $E_F$, $E_{BG}$ and $E_{DS}$ are the Fermi level, the electric field generated by the gate electric pulse and the electric field generated by the bias voltage, respectively. The blue, orange and black arrows represent the direction of electron tunneling, local Electric Field and electronic transition, respectively. The "hν" represents radiant energy of light. h stands for Planck's constant and ν is the frequency of light. Source data are provided as a Source Data file (https://doi.org/10.6084/m9.figshare.25930456).

performance provides a solid foundation for infrared image detection, memory and computation.

Next, we further elucidated the working mechanism of PMC devices. The Kelvin Probe Force Microscope (KPFM) for material interface barriers is shown in Supplementary Fig. 5. Upon the application of a positive pulse $V_{BG}$, an electric field $E_{BG}$ is generated inside the device, as illustrated in Fig. 4d. Under this condition, a portion of the electrons produced by ionization within the channel undergoes

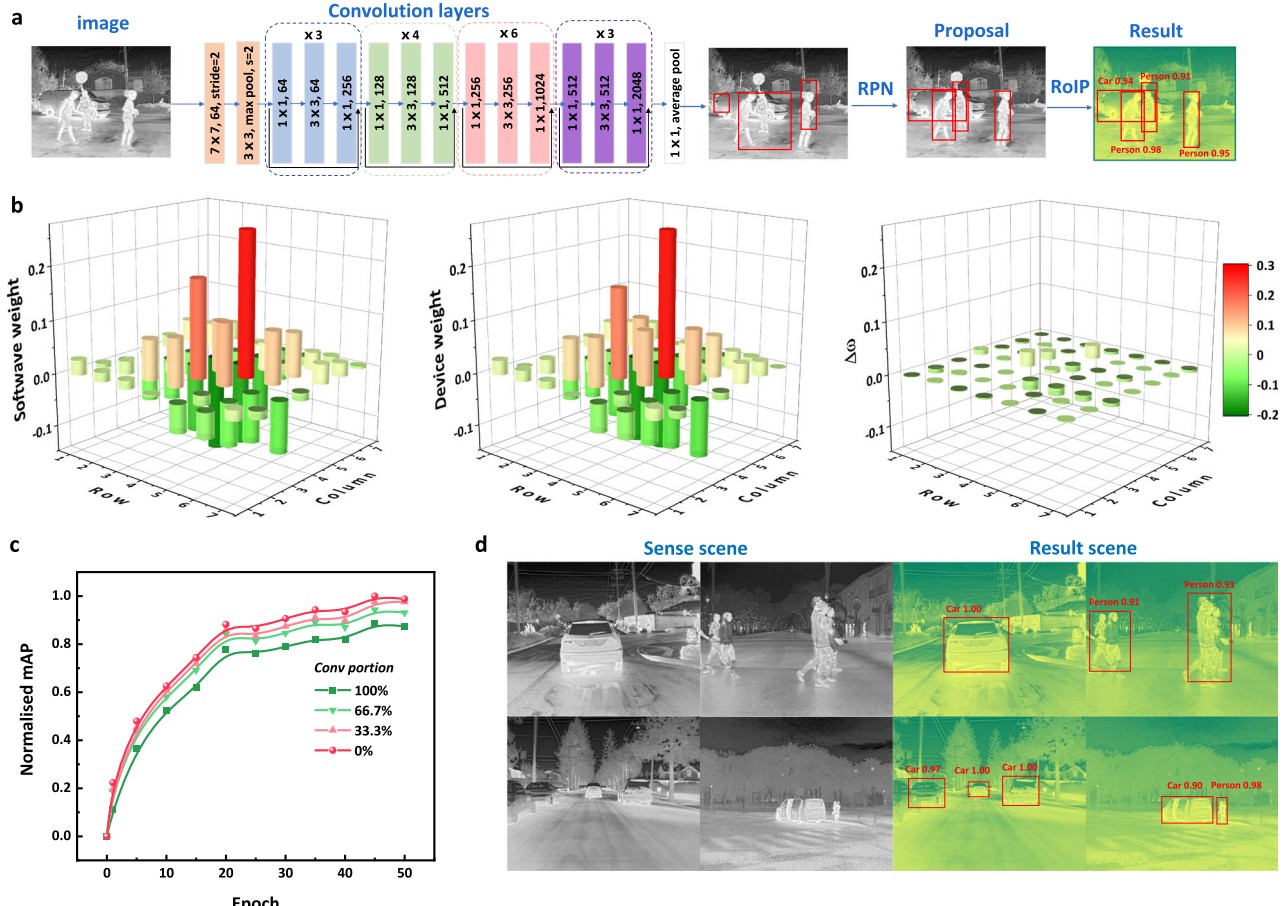

**Fig. 5 | Object detection and recognition based on MWIR scene. a** Image processing flow and neural network details Faster-RCNN (Fast Region-based Convolutional Network). RPN represents region proposal networks, Proposal represents the image with proper anchors. RoIP (Region of Interest Pooling) is a process of pooling. And ResNet50 is used as the convolution layer. **b** Represent software trained weight, device substitution weight and the weight error ($\Delta\omega$), respectively (one kernel of $7 \times 7$ array is shown as an example). Weight values are also listed in tabular form in Supplementary Tables 3 and 4. **c** The mean average precision (mAP) of different portion (0%, 33.3%, 66.7%, and 100%.) of convolutional layer substitutions. **d** Grayscale image on the left panel represents the image of midwave infrared sensing. Corresponding recognition results and accuracy are displayed on the right panel. Source data are provided as a Source Data file (https://doi.org/10.6084/m9.figshare.25930456).

tunneling through the $h$-BN layer into the graphene layer via Fowler-Nordheim (FN) tunneling[21], while the holes remain within the channel. The lower electron concentration in the channel results in a lower conductance state based on $\sigma = nq\mu$ ($\sigma$: conductance, $n$: electron concentration, $\mu$: electron mobility). Firstly, from the photomapping diagram (see in Supplementary Fig. 24), the photocurrent is mainly generated by photogenerated electron-hole pairs in the depletion region of the heterojunction under the action of the built-in electric field. The photogenerated carriers come from the covalent electrons between boron and phosphorus atoms in the depletion region of the heterojunction under mid-wave infrared excitation. In the depicted low-conductance state shown in Fig. 4e, the lower concentration of electrons within $MoS_2$ leads to a narrower width of the BP depletion region, reducing the number of covalent bonds that can be excited and thereby weakening the photocurrent. In Fig. 4f, with the application of a negative pulse $V_{BG}$ is applied, an opposite electric field $E_{BG}$ is generated inside the device. Part of the electrons from the graphene conductive band tunnel through the $h$-BN layer into the heterojunction channel layer, influenced by the electric field. The higher electron concentration in the channel corresponds to a higher conductance state. In the illustrated high-conductance state in Fig. 4g, the higher electron concentration in $MoS_2$ leads to the widening of the BP depletion region, so that the number of covalent bonds that can be

excited becomes larger, resulting in the enhancement of the photocurrent.

## Object detection and recognition based on MWIR scene

The structure of the Faster-RCNN neural network[28,29] used to distinguish objects intelligently contains four main parts: convolution layers, region proposal network (RPN), Region of Interest Pooling (RoIP), and classification. The ResNet50 is utilized as the convolution layer, and the corresponding structure is depicted in Fig. 5a. The image photodetectors receive two-dimensional image light information, specifically a two-dimensional mid-wave infrared light power array. MAC can be achieved by scanning spatial power points based on convolution kernel. Thus, the feature extraction function of convolution layer networks can be replaced with the performance of PMC devices. In the left panel of Fig. 5b, the 3D histogram of the trained $7 \times 7$ convolution kernel weight (taking one kernel for an example) is displayed. The response weight sequence of the device's blackbody radiation response at a wavelength of 3600 nm is extracted (see Supplementary Table 2). By linearly mapping the PMC device's responsivity to the trained weights, the device weights are obtained in the middle panel of Fig. 5b. The weight error of the $7 \times 7$ kernel, caused by discrete responsivity, is displayed in the right panel of Fig. 5b. The weight change of the above text is achieved through the modification of HDF5

**Table 1 | Comparison of key parameters with previously reported optoelectronic memories and in-sensor computing**

| Materials | Programming time (s) | dynamic range | Wavelength (nm) | R (A/W) | PPE (J) |
|---|---|---|---|---|---|
| Doped silicon[38] | External memory | / | 473 | / | / |
| WSe$_2$[19] | External memory | / | 650 | 0.05 | / |
| VO$_2$ films[39] | 1 | 1 | 650 | / | $10^{-7}$ |
| MoS$_2$[40] | $10^{-2}$ | <10 | 660 | / | $2 \times 10^{-9}$ |
| NbS$_2$/MoS$_2$[41] | 0.5 | $10^4$ | 532 | 1.1 | $3 \times 10^{-12}$ |
| MoS$_2$[42] | 10 | 10 | 520 | 0.033 | $1 \times 10^{-12}$ |
| MoTe$_2$[20] | $10^{-5}$ | $10^3$ | 1310 | 0.8 | $5 \times 10^{-14}$ |
| MoS$_2$ flash | $2 \times 10^{-8}$ | >$10^6$ | 520 | / | $4.5 \times 10^{-15}$ |
| BP flash | $2 \times 10^{-8}$ | $10^4$ | 1550 | 0.0125 | $1.8 \times 10^{-12}$ |
| PMC device | $2 \times 10^{-8}$ | >$10^6$ | 3900 (Blackbody) | 1.68 | $1.8 \times 10^{-15}$ |

*PPE* per-programming energy: minimum energy of distinguishable changing conductivity states.

files in Python language. Subsequently, mean average precision (mAP) (detailed in Supplementary Note 2) is introduced as an indicator to evaluate the accuracy of overall recognition. This evaluation is conducted using the test dataset every 5 epochs. In Fig. 5c, the normalized mAP gradually increases, with the increasing epoch in the front 50 epochs, where the last three parts of the Faster-RCNN are trained with the ResNet50 feature extraction network fixed. As the proportion of weight substitution in the convolutional layer increases from 0% to 100%, the mAP decreases to 89% of its initial value. This underscores that the device's performance can indeed substitute the role of the feature extraction network. The loss converging with epoch is displayed in Supplementary Fig. 34. We provide a brief diagram of the array circuit connection mode to achieve negative weight (taking the $3 \times 3$ [1, −1, 1; 1, 1, 1; 1, 1, 1] as an example), and detailed equivalent transistor connection can be found in Supplementary Fig. 31. The actual detection results are shown in Fig. 5d. The left panel displays the infrared detection image, while the right panel showcases the corresponding detection results, including the location, type and accuracy of the object. In those results, the accuracy of common car body recognition and person is all above 90%. Besides, image preprocessing can also be realized (seen in Supplementary Figs. 32 and 33). This lays the groundwork for the use of hardware weights as an alternative to convolutional neural networks.

## Discussion

In summary, we developed PMC devices with BP/MoS$_2$/$h$-BN/graphene vdWs stacked integration, showcasing ultrafast flash memory and MWIR sensing capabilities. Table 1 summarizes the memory characteristics and photoresponse characteristics of our PMC device and other reported optoelectronic memories. The PMC device facilitates low-power execution of the MAC operation of MWIR target detection and recognition. A pivotal feature of the PMC device is the capacity to store and modify responsivity, essentially representing the weights of neural networks. Firstly, with an ultrafast flash architecture (20 ns), the device has a low per-programming energy (PPE) of 1.8 fJ. Additionally, the PMC device achieves an extensive mid-wave infrared responsivity programming dynamic range of 1.68 A/W in 3600 nm under blackbody radiation, and the cut-off wavelengths of 3900 nm at room temperature. Based on the 49 μs (Supplementary Fig. 25) rising time of our PMC devices' photoresponse, the time of image recognition and processing is on the order of microseconds. It is faster than conventional CMOS image processing system which includes ADC data conversion time (10 ~ 50 μs)[30–32] and the time of image processing and recognition (~100 ms)[33]. In the future, optimizing the response time of the PMC device will make the processing time of the system even shorter (details compared to conventional image processing system with commercial uncooled MCT MIR detectors are displayed in Supplementary Table 5). The device weights effectively substitute

feature-extraction network weights in Faster-RCNN network simulations, achieving 89% of the software training mAP. With the advancements of large-scale production of materials[11,34,35], selective etching process and advanced transfer technology such as stamp transfer technology[36,37], it is very promising for large-scale preparation of PMC arrays. PMC hardware provides an efficient way to integrate infrared photodetection, memory and computing, laying the foundation for PMC-type devices to be used in advanced intelligent vision assisted driving systems.

## Methods

### Device fabrication

We selected SiO$_2$/Si (300 nm SiO$_2$ grown on p-doped Si substrates) as the substrate. In the initial step, the substrate underwent meticulous cleaning with acetone and ethylene glycol to ensure a smooth and flat surface. Subsequently, we transferred the 2D materials onto a fixed-point transfer platform. All 2D material flakes (BP, MoS$_2$, $h$-BN and graphene) were mechanically exfoliated using polydimethylsiloxane (PDMS) from bulk crystal (from HQ Graphene). Employing dry transfer technology, graphene, $h$-BN and MoS$_2$ were successively transferred to the substrate. The device materials on the substrate were annealed in a nitrogen atmosphere at 250 degrees Celsius for 2 h. Then, the BP was rapidly transferred onto the material stack to prevent excessive oxidation of BP. Next, the polymethy1 methacrylate (PMMA) (AR-662.09) polymer was applied to the surface of the device to isolate it from air and water. Finally, the Cr/Au (10 nm/70 nm) source (S) and drain (D) electrode patterns were defined by electron-beam photolithography (EBL) and deposited by electron-beam evaporation (EBE). To prevent BP from oxidation during measurements, the device surface was coated with PMMA and the electrode pads were selectively exposed using EBL.

### Material characterizations and device measurements

Atomic force microscopy (AFM) measurements for the devices were conducted with an MFP-3D Origin+ (Asylum Research, Oxford Instruments) system. Tapping mode was performed a conductive platinum-coated antimony (n) – doped silicon tip (SCM-PIC, Veeco) was used. A cross-sectional analysis was performed utilizing high-resolution transmission electron microscopy (HRTEM) technology. Furthermore, energy-dispersive X-ray spectroscopy (EDS) elements mapping was employed for comprehensive elemental analysis. Raman spectroscopy was employed to characterize the four materials: BP ($361 \, cm^{-1}$, $439 \, cm^{-1}$, $467 \, cm^{-1}$); MoS$_2$ ($385 \, cm^{-1}$, $404 \, cm^{-1}$); $h$-BN ($1375 \, cm^{-1}$); graphene ($1580 \, cm^{-1}$, $2700 \, cm^{-1}$). All the electrical characterization was carried out under atmospheric conditions in the dark at room temperature. Direct current signals were produced using the source/monitor unit in the B1500A. The voltage pulses were generated using the semiconductor pulse generator unit in the B1500A.

The waveform of the transient photocurrent response was checked using a DPO 5204 oscilloscope (Tektronix) after amplification by the preamplifier. Photocurrent mapping was obtained by laser scanning and ammeter reading, with the device positioned at the center of the light spot during measurements (see Supplementary Fig. 24). Optical stimulation was administered using light lasers with wavelengths of 638 nm, 1064 nm, 1310 nm, and 1550 nm (emitted and controlled by the Thorlabs), and corresponding light intensities were measured using a light intensity meter.

**Detection and recognition of infrared thermal traffic image**

The traffic recognition at night simulation was performed using Microsoft Visual Studio Code based on Python 3.7.16 (tensorflow2). Faster-RCNN is used as the neural network architecture (containing convolutional layers, region proposal networks, Roi-pooling, and classification layers). Training of the neural network was performed using the FLIR thermal image dataset. The backbone feature extraction network ResNet50 was initially imported for training the region proposal networks (RPN), ROI-pooling, and fully connected neural networks. During the training process, the convergent loss function and the HDF5 weight file for each epoch can be obtained. Subsequently, the response weight sequence of the device's blackbody radiation response at a wavelength of 3600 nm was extracted. This set of weights was linearly mapped to a set of convolution layers in the feature extraction network, requiring modifications to the HDF5 weight file in Python. Ultimately, the mAP index was used to compare and analyze whether the weight of the device's responsivity can effectively replace the weight trained by the computer.

## Data availability

The Source Data underlying the figures of this study are available at https://doi.org/10.6084/m9.figshare.25930456. All raw data generated during the current study are available from the corresponding authors upon request.

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

## Acknowledgements

This work was supported by the National Key Research and Development Program of China (Grant Nos. 2023YFB3611400, W.H.; 2021YFA1200500, P.Z), National Natural Science Foundation of China (Grant Nos. 61925402, P.Z; 62090032, P.Z; and 62304040, Y.W.), Science and Technology Commission of Shanghai Municipality (Grant No. 19JC1416600, P.Z.), China National Postdoctoral Program for Innovative Talents (Grant No. BX20220082, Y.W.), China Postdoctoral Science Foundation (Grant No. 2022M720750, Y.W.), Open Fund of State Key Laboratory of Infrared Physics (Grant No. SITP-NLIST-ZD-2023-01, Y.W.) and the Strategic Priority Research Program of the Chinese Academy of Sciences (Grant No. XDB44000000, P.Z.).

## Author contributions

Y.W., W.H., and P.Z. conceived the device idea and supervised the work. Y.W., and Y.Z. designed the experiments. Y.Z. conducted the theoretical analysis with the assistance of Y.W. and X.P. Y.Z. performed device fabrication and achieved the device performance. Y.Z. and Y.J. support the characterization of materials. Q.L. and Z.W. provided some assistance on blackbody and FTIR test. Y.Z. came up with and carried out the application simulation. X.L. and C.L. provided some instruction on simulation. Y.Z., Y.W., W.H., and P.Z. co-wrote the manuscript and all authors contributed to the discussion and revision of the manuscript.

## Competing interests

The authors declare no competing interests.
