## [Peer Review File · Nature Communications]

Non-volatile 2D MoS₂/black phosphorus heterojunction photodiodes in the near- to mid-infrared region

Corresponding Author: Dr Yang Wang

One or more attachments were originally included by the reviewers as part of their assessment. The content of those attachments are not included in this file.

Reviewer comments:

Reviewer #1

(Remarks to the Author)
Comments

In this work, Yuyan Zhu et al. demonstrate the PMC devices, which are used as ultrafast flash memory, and also show MWIR sensing capabilities in the mid-infrared range. Based on the charge tunneling in the graphene floating gate through the electrical gating method, the conductance, and photoresponsivity of the PMC can be precisely programmed. As a result, the demonstrated multifunctional PMC holds the promise to realize more complex CNNs for machine vision sensors distributed with edge computing. The paper is well organized, and the main results are convincing and interesting. Furthermore, I suggest that the authors revise the manuscript to address the issues discussed below.

1. In Figure 3a, a retention time of approximately 1000s is observed. Clarification on the methodology for determining this retention time is necessary. Additionally, an exponential fitting of the I_{ds} against time would enhance data analysis clarity. Furthermore, given the susceptibility of BP material to oxidation, it's crucial to evaluate potential degradation issues in the PMC device's performance. Were any signs of performance degradation observed during the operation? This aspect is critical for assessing the device's practical applicability.
2. Figure 3b, is there any specific reason for using a 1550 nm laser to test the ON/OFF stability of the device? As the title of the paper is "Non-volatile mid-wavelength infrared photodiodes", while mid-wavelength infrared should only cover the wavelength 3 μm to 5 μm in the definition.
3. In the manuscript, it appears that the authors operated the CNN on a physical PMC array with a size of 7 by 7. While this serves as a viable research prototype at the current stage, considering future technological advancements, it would be insightful to contemplate the feasibility of scaling up PMC arrays to larger sizes. Envisioning the fabrication of PMC arrays with larger array sizes presents challenges such as the growth of materials like BP and MoS₂ over larger areas, as well as the implementation of large-scale 2D flake transfer techniques. A brief discussion on this perspective in the concluding paragraph would enrich the scope of the work and provide valuable insights into potential future directions.
4. In the Supplementary Materials, the author demonstrates that the total dead time (including rise and fall times) of each device is 100 μs , thereby limiting the operational bandwidth of the PMC to 10 kHz. Given this limitation, it is pertinent to address how the PMC device could potentially accelerate signal processing compared to conventional CMOS techniques. A brief insights on this perspective in the concluding paragraph would be also helpful.
5. Determining the portion of parameters stored in PMC devices compared to those stored in digital computers is crucial for evaluating the efficacy of PMC utilization, particularly in the context of convolutional layers. If only a negligible portion of parameters is stored using PMCs, the comparative benefit of employing PMCs as the convolutional layer becomes limited. Therefore, a thorough analysis is necessary to quantify the proportion of parameters stored within PMC devices relative to those stored in digital computers.
6. In Figure 5b, the author plots both the weight and the weight error. It would be beneficial for the author to estimate the accuracy of the weight-setting process, particularly in terms of the number of bits of accuracy when programming the PMC.

This estimation would provide valuable insights into the precision of weight programming in the PMC. Addressing these considerations would contribute to a comprehensive understanding of the PMC's capabilities and broaden the potential applications in advanced neural network architectures.

7. Some of the labels in the figure are misleading and the figure captions are not well descriptive. For example, "Initially, a positive pulse (amplitude 30 V) is applied to the VBG input to ensure that the device is reset to the off-state with low conductance. Subsequently, it is programmed by a negative pulse (amplitude -30 V, width 20 ns). Varying the number of the VBG pulses allows the device to be programmed to more than 12 stable and distinguishable states with a long retention time exceeding 103 s". My understanding is the author employs a single positive pulse as the initial reset pulse to transition the device to a low-conductance off-state. Subsequently, a sequence of negative pulses is applied, with increasing pulse widths, to set the device to multiple states. that correct? In Fig. 3a, it is confusing to list the pulse parameters -30V, and 20 ns individually while keeping other pulse parameters in the box. Also, in Figure 3c, the parameter "Vlight" is not given anywhere else in the manuscript. Also, in the middle panel, the "20 ns" are confusing and not matched with the time scale label in the bottom panel from "17 ps" to "1.1 μs".

In conclusion, in its present form, there are several flaws and missing information that need to be addressed before considering publication in Nature Communications.

Reviewer #2

(Remarks to the Author)

In this manuscript, the authors present a non-volatile mid-infrared photodiode constructed using a two-dimensional graphene/h-BN/MoS₂/BP semi-floating gate heterostructure. The device architecture incorporates graphene/h-BN/MoS₂ as a flash memory cell and MoS₂/BP as a photodetector, enabling programmable conductance states and responsivity via a pulsed gate with a pulse width of 20 ns. These states can be retained even after removing the gate, allowing for the detection, memory, and computing of mid-infrared information. While the device architecture and functional demonstration are interesting, several concerns regarding its working mechanism and practical applications warrant further investigation. Therefore, I cannot recommend its publication at the current stage.

1. The device's response wavelength is limited to 4 μm, aligning with the blackbody radiation spectrum at 1000 K. However, real-world scenarios, such as the traffic scene mentioned by the authors, typically involve environmental objects radiating at wavelengths around 10 μm at room temperature. This limitation suggests that the device's practical utility requires validation.
2. The FLIR thermal image dataset and the device's responsivity were used to simulate object detection and recognition. Typically, thermal imagers detect radiation wavelengths from room-temperature objects in the range of 7-14 μm. However, the device's detection range is limited to 4 μm. The authors attempted to use the responsivity to the short wavelength range to simulate information from the long wavelength range, which is not rigorous. Therefore, the claim of achieving a 100% recognition rate for cars lacks credibility.
3. Comparatively, uncooled thermal imagers coupled with CNN algorithms demonstrate superior performance in detecting and recognizing infrared thermal traffic images across low, room, and high-temperature environments. This existing solution appears more versatile than the proposed device's application.
4. The manuscript highlights an ultrafast (20 ns) weight update process, reducing weight programming power consumption to 1.8 fJ. Although emphasized, the discussion lacks depth. Further exploration into how pulse width affects channel conductance and the factors constraining minimum pulse width is necessary.
5. A lower conductance results in a lower responsivity, which was ascribed to the reduced number of ionizable atoms within the heterojunction channel. This explanation lacks clarity. Defining the number of ionizable atoms and considering interface barriers, especially those of MoS₂-BP and MoS₂-metal, is essential for comprehensively explaining the device's operation. More experimental supports are recommended.
6. The device's infrared light response predominantly arises from BP, while the differing dependence of near-infrared and mid-infrared responsivity on the conductance of the device remains unexplained. What is the underlying reason? What impact will this difference have on subsequent computing?
7. Material thickness significantly impacts device performance, as noted by the authors. However, the criteria for determining the thickness used in the study are undisclosed. Comprehensive experimentation exploring the effects of various thicknesses on performance is advisable.
8. What is the mid-infrared detectivity of the device? What is the influence of various conductance on detectivity?
9. Several minor issues are noted:
 - 9.1 Reference No. 26 cited in the manuscript is unrelated to the content. "The photocurrent spectrum of the device in the MWIR band was obtained by Fourier transform infrared spectroscopy (FTIR)²⁶." "26. Li, T. et al. Developing fibrillated cellulose as a sustainable technological material. 364 Nature 590, 47-56 (2021)."
 - 9.2 Most reference citations come from the authors' papers. It is recommended to broaden the reference sources.
 - 9.3 The photocurrent mapping was mentioned in the Methods section. However, the corresponding results can't be found in the manuscript.

Reviewer #3

(Remarks to the Author)

The manuscript describes a novel Van der Waals (VdWs) photodetector for Mid-wave infrared (MWIR) detection, which fully demonstrates the unique properties of VdWs materials. The work is very interesting and will further promote the application

of VdWs materials for MWIR detection and recognition with heterojunction and high performance. Among the previously proposed works, devices used for multiply-accumulate operations require additional memory to store weights and do not allow for efficient functional integration. Zhu et al in this paper realized the in-sensor computing hardware that integrates MWIR photodetector with adjustable responsivity, memory, and computing simultaneously. The device has the potential to realize recognition of objects at night through MWIR detection with lower programming energy consumption. The results are convincing in the sense that they provide sufficient evidence that MWIR photodetector has non-volatile memory characteristics including dynamic range, maintenance, and durability. The authors also measured the linear correspondence between different conductance and responsivity in real-time. The measurement of blackbody radiation proved the possibility of practical use of the device. Hence, this reviewer would like to recommend the paper for publication in Nature Communications if the authors could address the following questions:

1. The ultrafast 20 ns pulse generated by the B1500 is usually not an ideal square wave. The authors should provide the actual pulse applied on the device through the oscilloscope.
2. The amplitude and pulse width of the pulses are expected to have varying effects on the memory state. To comprehensively understand the pulse modulation effectiveness, it is recommended that the authors provide a range of pulse modulations with different amplitudes and pulse widths.
3. Why the device structure should be a semi-floating gate and what will happen if it is stacked directly vertically. The authors need to provide test results for direct stacking.
4. The position of photocurrent generation is very helpful in exploring the mechanism of photocurrent, and the authors should provide photocurrent mapping images at a specific wavelength.
5. It has been demonstrated in the paper that the device has a stable responsivity under irradiation with different power densities of 1550 nm laser. The authors need to provide test results at 1310 nm to demonstrate the consistency of the conclusions.
6. In Figure 5b, a small difference occurs in the responsivity replacement software weights of the devices. The author should add the weight values in tabular form to make it clearer.

Author Rebuttal letter:

[*The author's responses to these comments can be found at the end of this file.*]

Version 1:

Reviewer comments:

Reviewer #1

(Remarks to the Author)

Comments from reviewer:

I have carefully reviewed the authors' responses and am pleased to note that most of my previous questions have been comprehensively addressed. The provided information is clear, detailed, and meets the expected standards. However, I still have a significant concern regarding the use of the reported PMC devices as the black radiation detectors in the memory CNN described by the authors. For a device to be effective in this context, it is critical that its responsivity remains linearly proportional to the stored conductance. Deviations from this linearity can lead to increased computation time and reduced efficiency. In the MWIR or black radiation task presented in the revised manuscript, the device exhibits a nonlinear behavior, which is nonideal for CNN applications within the MIR range. This nonlinearity undermines the motivation for using the PMC device for MWIR object detection tasks, as it can compromise performance and computational efficiency.

I believe the authors should address this issue and provide a resolution for the nonlinearity of the PMC device's responsivity before the manuscript can be considered for publication in Nature Communications.

Reviewer #2

(Remarks to the Author)

The authors have addressed several of my concerns; however, two issues remain unclear.

First, the authors stated that the overall time of conventional image processing systems exceeds 100 ms. This includes the photoresponse time of an uncooled mid-wave imager (100 μ s to 25 ms), ADC data conversion time (approximately 10-50 μ s), and the reasoning time for image processing and recognition using computer software (at least 100 ms). A key point is that the photoresponse time of commercial uncooled MCT MIR detectors can reach the order of nanoseconds, which is significantly faster than the presented PMC devices (49 μ s). Another key point is that image processing and recognition with the presented PMC devices still require the assistance of computer software, necessitating a reasoning time of at least 100 ms. Therefore, the PMC device does not exhibit any speed advantage. It is suggested to provide a clearer comparison between the PMC device and conventional image processing systems.

Second, the responsivity of the PMC device at 3600 nm is 1.68 A/W (Fig. 4b), while the maximum responsivity at 3622 nm is lower than 0.28 A/W (Supplementary Fig. 26d). How should this difference be understood?

Reviewer #3

(Remarks to the Author)

The authors have addressed all the concerns and this reviewer would like to recommend this revised manuscript for publication in Nature Communications.

Author Rebuttal letter:

[*The author's responses to these comments can be found at the end of this file.*]

Version 2:

Reviewer comments:

Reviewer #1

(Remarks to the Author)

I think the authors have addressed my comments. The information is clear, comprehensive, and aligns well with my expectations. As such, I think this work is worth to be published on nature communications

Reviewer #2

(Remarks to the Author)

The authors have addressed my concerns and I recommend publishing this revised version on Nature Communications.

Author Rebuttal letter:

Responses to Reviewers's Comments

Thank the reviewers for carefully reviewed our manuscript "Non-volatile 2D MoS₂/black phosphorus heterojunction photodiodes in the near- to mid-infrared region" (NCOMMS-24-12907B). Here's the point-by-point response to their comments with the response marked in blue and the comments in black.

Reviewer #1 (Remarks to the Author):

I think the authors have addressed my comments. The information is clear, comprehensive, and aligns well with my expectations. As such, I think this work is worth to be published on nature communications

Response: We thank the referee for recommending the publication of our work in Nature Communications.

Reviewer #2 (Remarks to the Author):

The authors have addressed my concerns and I recommend publishing this revised version on Nature

Communications.

Response: We thank the referee for recommending the publication of our work in Nature

Communications.

1

Responses to Reviewers' Comments

We acknowledge the reviewers for carefully reading our manuscript '*Non-volatile mid-wavelength infrared photovoltaic detectors* (NCOMMS-24-12907)' and providing constructive comments on our work. According to the reviewers' comments, we have carefully revised our manuscript and provided more detailed data to improve the manuscript's readability. We reorganized the manuscript and Supplementary Information to present the new data with elaborate discussions and abundant experiment data. With the help of the reviewers, we believe that the revision has been significantly improved. To clearly state our research, new discussions have been added in the revision to make our findings more solid. The corresponding revisions concerning comments have been provided and highlighted in red in the revised manuscript and Supplementary Information. The corresponding responses of the reviewers are marked by blue words. The detailed revisions in the revised manuscript are listed on a separate page at the end of the response letter.

Responses to Reviewer # 1:

General Comments:

In this work, Yuyan Zhu et al. demonstrate the PMC devices, which are used as ultrafast flash memory and also show MWIR sensing capabilities in mid infrared range. Based on the charge tunneling in the graphene floating gate through the electrical gating method, the conductance, and photoresponsivity of the PMC can be precisely programmed. As a result, the demonstrated multifunctional PMC holds the promise to realize more complex CNNs for machine vision sensors distributed with edge computing. The paper is well organized, and the main results are convincing and interesting. Furthermore, I suggest that authors revise the manuscript to address the issues discussed below.

Response:

Thank the reviewer for carefully reading the manuscript and positive comments on our work 'The paper is well organized, and the main results are convincing and interesting'. Meanwhile, the reviewer has concerns about the potential degradation, response time, and neural network simulation. Based on these critical and constructive

comments, the new experimental data have been added to the revised manuscript to enhance the readability. With the help of the reviewer, the whole manuscript has been largely improved. In the following, we will address all comments point-by-point and revised the manuscript. We hope that the revised manuscript would remove the reviewers' concerns.

Comment 1:

In Figure 3a, a retention time of approximately 1000s is observed. Clarification on the methodology for determining this retention time is necessary. Additionally, an exponential fitting of the I_{ds} against time would enhance data analysis clarity. Furthermore, given the susceptibility of BP material to oxidation, it's crucial to evaluate potential degradation issues in the PMC device's performance. Were any signs of performance degradation observed during the operation? This aspect is critical for assessing the device's practical applicability.

Response:

Thank the reviewer for the valuable comment. We appreciate your careful consideration and are grateful for the opportunity to address your concerns. Clarification on the methodology for determining this retention time is indeed necessary.

The loss of the conductance state is mainly due to the leakage of the captured charge by tunneling under weak fields, which is thus manifested by a change in the threshold voltage. Here we define the conductance state failure criterion as a 50% loss of captured charge, in other words, the time corresponding to the shift of the threshold voltage difference to 50% of the initial value is the conductance retention time. In the **Fig. 3a** of the previous manuscript, we presented the retention time of approximately 1000s. To further elucidate the retention performance, we conducted extensive measurements of the retention performance based on the shift of the threshold voltage, as depicted in **Fig. R1.1a-c**. These results further clarify the methodology for determining retention time of our device. Device data can be maintained for several years with this test method. In **Fig. R1.1d**, we also performed exponential fitting of I_{ds} against time using the function Asymptotical ($y=a-b*c^x$) for both low resistance state and high resistance state (LRS and HRS). we make an exponential fitting of the I_{ds} against time. The exponential fit function is Asymptotical ($y=a-b*c^x$). LRS ($a=1.56E-$

12, $b=-7.37E-13$, $c=0.99794$); HRS ($a=1.02E-5$, $b=-3.28E-6$, $c=0.99939$). From the results of the exponential fit, we estimate that the high and low resistance states of the device can be held for up to 10 years.

In consideration of the susceptibility of BP material to oxidation, we coated a layer of PMMA on the surface of the device to insulate it from moisture and oxygen. We show the original memory characteristics of PMC devices in **Fig. R1.2**. As a comparison, we show the memory characteristics of PMC devices after 15 days in **Fig. R1.3**. The results show that the PMMA coating can effectively reduce the effect of BP oxidation on the device performance.

Fig. R1.1 | Verification of non-volatile retention characteristics. **a**, I_{DS} - V_{BG} hysteresis curve of the PMC device with $V_{DS} = 1V$, **b**, Transfer characteristic curves of the PMC device (after a V_{BG} pulse= -30 V for 50 ns duration erasing operation) at a different time interval. The original state_0 was achieved by 30 V for 1 s pulse. **c**, Retention-failure rates for the device. **d**, Exponential fitting of the I_{DS} against time. From the results of the exponential fit, the high and low states of the device can be held for up to 10 years.

Fig. R1.2 | Memory characteristics of pristine PMC devices. a, b, I_{DS} - V_{DS} curve of the PMC device-1 with different V_{BG} . I_{DS} - V_{BG} hysteresis curve of the PMC device-1 with $V_{DS} = 1$ V. **c, d,** I_{DS} - V_{DS} curve of the PMC device-2 with different V_{BG} . I_{DS} - V_{BG} hysteresis curve of the PMC device-2 with $V_{DS} = 1$ V.

Fig. R1.3 | Memory characteristics of PMC devices after 15 days. a, b, I_{DS} - V_{DS} curve of the PMC device-1 with different V_{BG} . I_{DS} - V_{BG} hysteresis curve of the PMC device-2 with $V_{DS} = 1$ V. **c, d,** I_{DS} - V_{DS} curve of the PMC device-2 with different V_{BG} . I_{DS} - V_{BG} hysteresis curve of the PMC device-2 with $V_{DS} = 1$ V.

Thank the reviewer for the constructive comments again, which helped us further test the extra characteristics of the devices. Moreover, your insightful suggestions not only guided us in testing the additional characteristics of our devices but also deepened our understanding of the memory methodology.

Revision:

We have reorganized **Supplementary Fig. 15-17** to display the durable properties of the device (**Supplementary, Page 19-21**). Additionally, corresponding discussions have been incorporated into the revised manuscript as follows: *“Additionally, we make a clarification on the methodology for determining the retention time and performed exponential fitting of I_{ds} against time, which shows that the high and low resistance states of the device can be held for up to 10 years. Furthermore, the degradation of the device can be effectively curbed by coating the BP materials surface with a passivation layer of PMMA (**Supplementary Fig. 15-17**).”*. (Page 10)

Comment 2:

Figure 3b, is there any specific reason using 1550 nm laser to test the ON/OFF stability of the device? As the title of the paper is “Non-volatile mid-wavelength infrared photodiodes”, while mid-wavelength infrared should only cover the wavelength 3 μm to 5 μm in definition.

Response:

Thank the reviewer for the valuable comment. In response, we have conducted tests on the device for electrical ON/OFF stability without laser illumination. This data is primarily used to demonstrate that the device can efficiently switch states without damage after 10^4 times of conductance state changes. The note for the light information was missing from the picture and has now been added.

The 1550 nm wavelength represents the longest wavelength at which our experimental instrument, the B1500A, in combination with a laser, can perform real-time electrical and optical testing of the device, as depicted in **Fig. 3c**. However, for testing the mid-wave infrared response, a separate mid-wave infrared laser is utilized, which is independent of the B1500A. Prior to examining the mid-wave infrared response at different conductance states, a pulse applied with the B1500A is necessary

to alter the device's conductance state, as illustrated in Figures 4a-c of the previous manuscript.

Besides, taking into account the reviewer's suggestion regarding the definition of "mid-wavelength infrared," as well as the experimental findings that PMC devices effectively integrate short-wave and mid-wave infrared with electrical memory properties, it would be appropriate to consider revising the title to "Non-volatile photodiodes in the near- to mid-infrared region" to accurately reflect the device's capabilities and the scope of the study.

Revision:

Combining the reviewers' suggestions with objective experimental results, we revised the title of the manuscript to “*Non-volatile photodiodes in the near- to mid-infrared region*”. (Page 1)

Comment 3:

In the manuscript, it appears that the authors operated the CNN on a physical PMC array with a size of 7 by 7. While this serves as a viable research prototype at the current stage, considering future technological advancements, it would be insightful to contemplate the feasibility of scaling up PMC arrays to larger sizes. Envisioning the fabrication of PMC arrays with larger array sizes presents challenges such as the growth of materials like BP and MoS₂ over larger areas, as well as the implementation of large-scale 2D flake transfer techniques. A brief discussion on this perspective in the concluding paragraph would enrich the scope of the work and provide valuable insights into potential future directions.

Response:

We appreciate the valuable insights provided by the reviewer regarding the scalability of PMC arrays to larger sizes. Indeed, considering future technological advancements, such as the large-scale growth of *h*-BN, MoS₂ and BP come true, it becomes feasible to envision the fabrication of PMC arrays with larger array sizes. To address the challenge of scaling up PMC arrays, we propose a potential fabrication approach. As shown in **Fig. R1.4**, firstly, the process involves transferring the *h*-BN/Graphene film onto a SiO₂/Si substrate, followed by array formation through the

etching technique. Subsequently, By using large-scale 2D flake transfer techniques such as stamp transfer technology^{1, 2}, we can transfer MoS₂ and BP flakes onto the corresponding positions in the array. By implementing these techniques, PMC arrays with larger sizes can be successfully realized.

In conclusion, while we operated the CNN on a physical PMC array with a size of 7 by 7, envisioning the scalability of PMC arrays to larger sizes opens up exciting possibilities for future research. By overcoming challenges associated with material growth³⁻⁵ and fabrication techniques, PMC arrays could potentially find applications in various fields, ranging from photonics to neuromorphic computing.

Fig. R1.4 | Schematic of PMC device array.

Revision:

We have added some discussion of the feasibility of scaling up PMC arrays to larger sizes in the concluding paragraph. Additionally, corresponding discussions have been incorporated into the revised manuscript as follows: “*With the advancements of large-scale production of materials, selective etching process and advanced transfer technology such as stamp transfer technology, it is very promising for large-scale preparation of PMC arrays.*”. (Page 16)

Reference:

1. Liu, G. et al. Graphene-assisted metal transfer printing for wafer-scale integration of metal electrodes and two-dimensional materials. *Nature Electronics* **5**, 275-280 (2022).
2. Nakatani, M. et al. Ready-to-transfer two-dimensional materials using tunable adhesive force tapes. *Nature Electronics* **7**, 119-130 (2024).
3. Wu, Z. et al. Large-scale growth of few-layer two-dimensional black phosphorus. *Nature Materials* **20**, 1203-1209 (2021).
4. Gao, T. et al. Temperature-triggered chemical switching growth of in-plane and vertically stacked graphene-boron nitride heterostructures. *Nature Communications* **6**, 6835 (2015).
5. Chen, S. et al. Wafer-scale integration of two-dimensional materials in high-density memristive crossbar arrays for artificial neural networks. *Nature Electronics* **3**, 638-645 (2020).

Comment 4:

In the Supplementary Materials, the author demonstrates that the total dead time (including rise and fall times) of each device is 100 μs , thereby limiting the operational bandwidth of the PMC to 10 kHz. Given this limitation, it is pertinent to address how the PMC device could potentially accelerate signal processing compared to conventional CMOS techniques. A brief insight on this perspective in the concluding paragraph would be also helpful.

Response:

We appreciate your valuable comment regarding the response time of the detectors and their implications for signal processing speed. In conventional CMOS image processing systems, ignoring the time of transmission delay and memory delay within the system, the overall system time primarily consists of the detector response time, ADC data conversion time and image processing and recognition time. Typically, CMOS detector response time is on the order of nanoseconds^{1, 2}, while ADC data conversion time averages about 10 to 50 μs ³⁻⁵. Moreover, in **Fig. R1.5**, the reasoning time for processing and recognition per image, based on computer software, is on the order of milliseconds⁶ (for example, Faster-RCNN inference time per image is at least 100 ms).

In contrast, PMC devices integrate photodetection, memory and processing functions. Through the connection of the device array in convolution topology circuits, the current generated by the array is the result of sensing the external picture for convolution calculation. It can also be understood that the quantization process of the ADC is transformed into the programming of the device's responsivity state. When

PMC arrays can be efficiently manufactured, the time of image recognition and processing mainly depends on the photoresponse time of the PMC device. The response time of our devices is on the order of microseconds. Consequently, the time required for image recognition and processing is significantly reduced compared to conventional CMOS techniques.

Looking ahead, optimizing the response time of PMC devices holds promise for further enhancing processing speed. Future advancements in PMC technology, including efficient manufacturing of PMC arrays, will contribute to even shorter processing times, ultimately revolutionizing signal processing applications.

Fig. R1.5 | Speed of Image Inference⁶. **a**, Accuracy vs time. **b**, GPU time (milliseconds) for each model.

Revision:

We have added some discussion of photoresponse times in the concluding paragraph. Additionally, corresponding discussions have been incorporated into the

revised manuscript as follows: “Based on the 49 μ s rising time of our PMC devices’ photoresponse, the time required for image recognition and processing is on the order of microseconds. It is faster than conventional CMOS image processing systems, which include ADC data conversion time (10~50 μ s) and the time of image processing and recognition (~100 ms). In the future, optimizing the response time of the PMC device will make the overall processing time of the system even shorter.”. (Page 16)

Reference:

1. Ojefors, E., Pfeiffer, U. R., Lissauskas, A. & Roskos, H. G. A 0.65 THz Focal-Plane Array in a Quarter-Micron CMOS Process Technology. *IEEE Journal of Solid-State Circuits* **44**, 1968-1976 (2009).
2. Bablich, A. et al. High-speed nonlinear focus-induced photoresponse in amorphous silicon photodetectors for ultrasensitive 3D imaging applications. *Scientific Reports* **12**, 10178 (2022).
3. Nie, K., Zha, W., Shi, X., Li, J., Xu, J. & Ma, J. A Single Slope ADC With Row-Wise Noise Reduction Technique for CMOS Image Sensor. *IEEE Transactions on Circuits and Systems I: Regular Papers* **67**, 2873-2882 (2020).
4. Park, H., Yu, C., Kim, H., Roh, Y. & Burm, J. Low Power CMOS Image Sensors Using Two Step Single Slope ADC With Bandwidth-Limited Comparators & Voltage Range Extended Ramp Generator for Battery-Limited Application. *IEEE Sensors Journal* **20**, 2831-2838 (2020).
5. Zhang, Z., Jiang, J., Chen, C. & Huang, X. The Design of A 12-Bit Two-Step Single Slope ADC For Carbon Based Image Sensors. 2023: IEEE.
6. Huang, J. et al. Speed/Accuracy Trade-Offs for Modern Convolutional Object Detectors. 2017: IEEE.

Comment 5:

Determining the portion of parameters stored in PMC devices compared to those stored in digital computers is crucial for evaluating the efficacy of PMC utilization, particularly in the context of convolutional layers. If only a negligible portion of parameters is stored using PMCs, the comparative benefit of employing PMCs as the convolutional layer becomes limited. Therefore, a thorough analysis is necessary to quantify the proportion of parameters stored within PMC devices relative to those stored in digital computers.

Response:

We appreciate the valuable comment from the reviewer and acknowledge the importance of determining the portion of parameters stored in PMC devices compared to those stored in digital computers, especially in the context of convolutional layers. In response to this concern, we have included additional data in the revised manuscript to provide a thorough analysis.

The PMC device has a combination of photodetection, storage and calculation functions, serving as a hardware alternative to the convolution layer. PMC integrated devices can directly complete the conversion and convolution of the operation. As the reviewer noted, the convolutional layer is crucial for achieving transformation, making it the target layer for device replacement. To quantify the proportion of substitution, we extended the range of convolutional layer substitutions and analyzed the proportion with 0%, 33.3%, 66.7%, and 100% substitution in **Fig. R1.6**. By quantifying the proportion of parameters stored within PMC devices relative to those stored in digital computers, the reviewer provides valuable insights into the comparative benefits of employing PMCs as the convolutional layer. We believe these findings enrich the scope of our work and contribute to a better understanding of the potential applications of PMC technology.

Fig. R1.6 | The mean Average Precision (mAP) – epoch curves under different weight profiles (0%, 33.3%, 66.7%, 100%).

Revision:

We have modified **Fig. 5d** to display the relative value of mAP with different weight replacement ratios (Page 29). Corresponding discussions have been incorporated into the revised manuscript as follows: “As the proportion of weight substitution in the convolutional layer increases from 0% to 100%, the mAP decreases to 89% of its initial value.”. (Page 15)

Comment 6:

In Figure 5b, the author plots both the weight and the weight error. It would be beneficial for the author to estimate the accuracy of the weight setting process, particularly in terms of the number of bits of accuracy when programming the PMC. This estimation would provide valuable insights into the precision of weight programming in the PMC. Addressing these considerations would contribute to a comprehensive understanding of the PMC's capabilities and its broaden the potential applications in advanced neural network architectures.

Response:

Thank the reviewer for the valuable comment. To estimate the accuracy of the weight setting process, we extracted the response weight sequence of the device's blackbody radiation response at a wavelength of 3600 nm, which is displayed in **Table R1.1**. In this table, along with the zero responsivity state, there are 22 states, resulting in an effective number of bits (ENOB) of 4.46 bits. If the device is operated by applying different constant gate voltages, then the device can traverse all states in the dynamic range. While this improves accuracy, it also results in large static power consumption and loses the advantage of non-volatility. Therefore, increasing the number of device states at a constant pulse is an effective way to improve accuracy while maintaining power consumption. Adjusting the programming pulse changes the number of device states. Pulse modulation under various pulse sets is displayed in **Fig. R1.7**. **Table R1.2** shows corresponding accuracy parameters.

Table R1.1. Response weight sequence (A/W).

0.41517	0.4351	0.46831	0.50152	0.52145	0.53806	0.55466
0.60781	0.62441	0.69416	0.7473	0.86687	1.19568	1.38832
1.42153	1.43814	1.47468	1.54442	1.59424	1.61417	1.70053

Fig. R1.7 | Pulse modulation under various pulse set. a-h, Pulse modulation results

under pulse set (-15 V, 50 ns), (-15 V, 100 ns), (-17 V, 100 ns), (-18 V, 100 ns), (-18 V, 50 ns), (-19 V, 100 ns), (-19 V, 50 ns) and (-20 V, 100 ns), respectively.

Table R1.2. Pulse modulation results.

V_{GS} and Pulse time	States number	Dynamic range (μA)	average conductance interval (μA)
-15 V, 50 ns	52 (5.7bits)	0.16	0.003
-15 V, 100 ns	30 (4.9bits)	0.56	0.0187
-17 V, 100 ns	27 (4.7bits)	1	0.037
-18 V, 100 ns	15 (3.9bits)	0.96	0.064
-18 V, 50 ns	22 (4.6bits)	0.4	0.0182
-19 V, 100 ns	10 (3.3bits)	1	0.1
-19 V, 50 ns	33 (5.0bits)	0.7	0.0212
-20 V, 100 ns	8 (3bits)	1.2	0.15

Revision:

We have added **Supplementary Table 2** to display the response weight sequence. (**Supplementary**, Page 41) And we have added **Supplementary Fig. 13** to show the pulse modulation under various pulse sets and **Supplementary Table 1** to summarize the effects of modulation in three ways. (**Supplementary**, Page 16-17). Corresponding discussions have been incorporated into the revised manuscript as follows: “Pulse modulation under various pulse sets can be found in **Supplementary Fig. 12**.”. (Page 10)

Comment 7:

Some of the labels in the figure are misleading and the figure captions are not well descriptive. For example, “Initially, a positive pulse (amplitude 30 V) is applied to the V_{BG} input to ensure that the device is reset to the off-state with low conductance. Subsequently, it is programmed by a negative pulse (amplitude -30 V, width 20 ns). Varying the number of the V_{BG} pulses allows the device to be programmed to more than 12 stable and distinguishable states with a long retention time exceeding 10^3 s”. My

understanding is the author employs a single positive pulse as the initial reset pulse to transition the device to a low-conductance off-state. Subsequently, a sequence of negative pulses is applied, with increasing pulse widths, to set the device to multiple states. that correct? In Fig. 3a, it is confusing to list the pulse parameter -30V, 20 ns individually while keeps other pulse parameters in box. Also, in Figure 3c, the parameter “V_{light}” is not given anywhere else in the manuscript. Also, in the middle panel, the “20 ns” are confusing and not matched with the time scale label in the bottom panel from “17 ps” to “1.1 μs”.

Response:

Thank the reviewer for the valuable comment. The reviewer's understanding of the detailed programming process for the conductance state is correct. In Fig. 3a, the parameters in the box represent the corresponding conductance states. Simens (S) is the unit of the conductance. We forgot to label the meaning of the parameter causing difficulties in understanding it. The same problem occurs in Figure 3c. “20 ns” is the pulse width and “17 pS” to “1.1 μS” represent the incremental conductances. **Fig. R1.8, 9** show the result of the change. In the meantime, we are sorry that we've lost the explanation of “V_{light}”. “V_{light}” is the drive voltage of the laser. Its amplitude is linearly related to the output power of the laser displayed in **Fig. R1.10**. We have carefully checked the whole manuscript and corrected the errors.

Fig. R1.8 | Multi-conductance states (1 pS to 16 μS) retention for 1000 s measured at VDS = 1 V.

Fig. R1.9 | Real-time test for conductance configuration and the photo-response at different conductance states (17 pS to 1.1 μS). “ V_{light} ” is the drive voltage of the laser. Its amplitude is linearly related to the output power of the laser. The laser pulse set (wavelength: 1550 nm, power density: $16 \mu\text{W}/\mu\text{m}^2$ ($V_{\text{light}} = 1.5 \text{ V}$), frequency: 1 Hz); The electrical pulse set (amplitude: -30 V, pulse width: 20 ns, frequency: 0.1 Hz).

Fig. R1.10 | Correspondence between 1550nm laser output power and laser drive voltage.

Revision:

We have reorganized **Figure. 3a and 3c** to better clarify the meaning of the parameters. (Page 27)

Responses to Reviewer # 2:

In this manuscript, the authors present a non-volatile mid-infrared photodiode constructed using a two-dimensional graphene/h-BN/MoS₂/BP semi-floating gate heterostructure. The device architecture incorporates graphene/h-BN/MoS₂ as a flash memory cell and MoS₂/BP as a photodetector, enabling programmable conductance states and responsivity via a pulsed gate with a pulse width of ~20 ns. These states can be retained even after removing the gate, allowing for the detection, memory, and computing of mid-infrared information. While the device architecture and functional demonstration are interesting, several concerns regarding its working mechanism and practical applications warrant further investigation. Therefore, I cannot recommend its publication at the current stage.

Response:

Thank the reviewer for carefully reading the manuscript and positive comments on our work ‘the device architecture and functional demonstration are interesting’. We appreciate your insightful comments on our research and understand that there are several concerns regarding the working mechanism and practical applications that need further investigation. We have revised the manuscript according to your suggestions and believe that these revisions have improved the paper. We believe these changes strengthen the manuscript and provide a clearer understanding of the device's working mechanism and its potential applications. With the help of the reviewer, the whole manuscript has been largely improved. In the following, we have addressed all comments point-by-point and revised the manuscript. We hope that the revised manuscript will remove the reviewers’ concerns.

Comment 1:

The device's response wavelength is limited to 4 μm , aligning with the blackbody radiation spectrum at 1000 K. However, real-world scenarios, such as the traffic scene mentioned by the authors, typically involve environmental objects radiating at wavelengths around 10 μm at room temperature. This limitation suggests that the device's practical utility requires validation.

Response:

Thank the reviewer for the valuable comment. As the reviewer says, "real-world scenarios typically involve environmental objects radiating at wavelengths around 10 μm at room temperature." In fact, the actual scene will also have some radiation intensity in the mid-wave infrared band at room temperature, which makes it possible for mid-wave infrared detection. The German companies Thermosensorik, FGAN-FOM reported in 2003 on their development of a dual-band infrared imaging system "CLEMENTINE" with two detectors¹. The mid-wave detectors are in the 3-5 μm band range. The image data captured by the system is shown in **Fig. R2.1**. In addition, the US Army Lockheed-Martin developed "MDSS Test" system with a 3.7-4.8 μm band range². In addition to these, there are many practical applications in single-detector dual-band imaging techniques. The U.S. Army Night Vision and Electronic Sensing Board, RVS (Raytheon Vision Systems), and OASYS, among others, reported in 2008 on the results of their third-generation thermal imaging camera validation prototypes (3rd Generation FLIR demonstrator)³. **Fig. R2.2** shows the image acquired with the prototype (middle wave: 3.6-5.3 μm). With the long-wave pixels approaching and exceeding the diffraction limit, dual-band imaging technology, which can make up for the lack of long-wave resolution by using medium-wave, will show greater advantages over long-wave single-band in terrestrial applications. Consequently, the device has practical application prospects and value⁴. With the rapid development of detector technology, low dark current and low noise will further improve the responsivity of MWIR detectors. This will enable PMC devices to perform well in room temperature imaging applications.

Fig. R2.1 | Images of buildings and roads at middle and long waves. a, b, c, Image of buildings, corresponding to middle wave, long wave and fusion. d, e, f, Image of buildings, corresponding to middle wave, long wave and fusion.

Fig. R2.2 | Dual-band images using the 3rd generation FLIR demonstrator of America³. a, Middle wavelength infrared. b, Long wavelength infrared.

In summary, we express our gratitude to the reviewers for their constructive comments. The comments on the limitation of device response bands allow us to explore and think more carefully about the feasibility of the practical application of the device, expanding the prospects and value of the device's application. Thank you for your thoughtful and constructive feedback.

Reference:

1. Schreer, O. et al. Helicopter-borne dual-band dual-FPA system. *Infrared Technology and Applications XXIX*; 2003: SPIE; 2003. p. 637-647.

2. Pollehn, H. K. & Ahearn, J. S. Multidomain smart sensors. *Infrared Technology and Applications XXV*; 1999: SPIE; 1999. p. 420-426.
3. King, D. F. et al. 3rd-generation MW/LWIR sensor engine for advanced tactical systems. *Infrared Technology and Applications XXXIV*; 2008: SPIE; 2008. p. 903-914.
4. Wen, M., Wei, L., Zhuang, X., He, D., Wang, S. & Wang, Y. High-sensitivity short-wave infrared technology for thermal imaging. *Infrared Physics & Technology* **95**, (2018).

Comment 2:

The FLIR thermal image dataset and the device's responsivity were used to simulate object detection and recognition. Typically, thermal imagers detect radiation wavelengths from room-temperature objects in the range of $\sim 7\text{-}14\ \mu\text{m}$. However, the device's detection range is limited to $4\ \mu\text{m}$. The authors attempted to use the responsivity to the short wavelength range to simulate information from the long wavelength range, which is not rigorous. Therefore, the claim of achieving a 100% recognition rate for cars lacks credibility.

Response:

Thank you for your valuable comments. According to our investigation, the FLIR dataset is a fusion of longwave infrared and visible light¹. We also learned that military detectors have a corresponding mid-wave infrared dataset for training, but the open source dataset lacks mid-wave infrared. In the past, in-sensor computing devices were mostly limited to the visible range, and the datasets used were mostly visible light datasets. Our device is the first to combine ultrafast storage with mid-wave infrared detection. Therefore, in order to fit the application scenario, we have selected the FLIR dataset from the open source dataset, which is a relatively good fit.

Whether it is an LWIR or MWIR dataset, the image is stored in the form of a grey scale map that can be processed by a computer. The difference between mid-wave infrared and long-wave infrared for the same scene is reflected in the difference in the grey values at the same positions of the grey scale map. This results in a difference in the training weights, and that is equivalent to the effect of changing an image. In fact, the accuracy of neural network substitution is mainly related to the dynamic range of the device and the number of conductance states².

Because the replacement ratio of convolutional layers is not large enough. The claim of achieving a 100% recognition rate for cars is not rigorous enough. In **Fig. R2.3**, we expand the proportion of convolutional layer substitutions to more rigorously judge

recognition accuracy. To quantify the proportion of substitution, we extended the range of convolutional layer substitutions and analyzed the proportion with 0%, 33.3%, 66.7%, and 100% substitution. The PMC hardware responsivity weights can reach 89% mean Average Precision index of the feature extraction network software weights.

Fig. R2.3 | The mean Average Precision (mAP) – epoch curves under different weight profiles (0%, 33.3%, 66.7%, 100%).

Revision:

We have modified **Fig. 5d** to display the relative value of The mean Average Precision (mAP) with different weight replacement ratios (Page 29). Corresponding discussions have been incorporated into the revised manuscript as follows: “*the PMC hardware responsivity weights can reach 89% mean Average Precision index of the feature extraction network software weights*”. (Page 2) “*As the proportion of weight substitution in the convolutional layer increases from 0% to 100%, the mAP decreases to 89% of its initial value.*”. (Page 15)

Reference:

1. Nirgudkar, S. & Robinette, P. Beyond visible light: Usage of long wave infrared for object detection in maritime environment. *2021 20th International Conference on Advanced Robotics (ICAR)*; 2021: IEEE; 2021. p. 1093-1100.
2. Chen, P. Y., Peng, X. & Yu, S. NeuroSim+: An integrated device-to-algorithm framework for benchmarking synaptic devices and array architectures. *2017 IEEE International Electron Devices Meeting (IEDM)*; 2017 2-6 Dec. 2017; 2017. p. 6.1.1-6.1.4.

Comment 3:

Comparatively, uncooled thermal imagers coupled with CNN algorithms demonstrate superior performance in detecting and recognizing infrared thermal traffic images across low, room, and high-temperature environments. This existing solution appears more versatile than the proposed device's application.

Response:

Thank the reviewer for the valuable comment. We will try to explain it from the point of view of the system speed advantage and the innovative development of the application.

(1) From the perspective of overall system speed

In conventional image processing systems, ignoring the time of transmission delay and memory delay within the system, the overall system time primarily consists of the detector response time, ADC data conversion time and image processing and recognition time. So far, the photoresponse time of uncooled mid-wave imagers is around $100\ \mu\text{s}\sim 25\ \text{ms}^{1-3}$. And ADC data conversion time averages about 10 to $50\ \mu\text{s}^{4-6}$. Moreover, in **Fig. R2.4**, the reasoning time for processing and recognition per image, based on computer software, is on the order of milliseconds⁷ (for example, Faster-RCNN inference time per image is at least 100 ms). So, the overall system time is over 100 ms.

In contrast, PMC devices integrate photodetection, memory and processing functions. Through the connection of the device array in convolution topology circuits, the current generated by the array is the result of sensing the external picture for convolution calculation. It can also be understood that the quantization process of the ADC is transformed into the programming of the device's responsivity state. When PMC arrays can be efficiently manufactured, the time of image recognition and processing mainly depends on the photoresponse time of the PMC device. The response time of our devices is on the order of microseconds ($49\ \mu\text{s}$). Consequently, the time required for image recognition and processing is significantly reduced compared to conventional techniques (with external memory and CNN algorithms).

Looking ahead, optimizing the response time of PMC devices holds promise for further enhancing processing speed. Future advancements in PMC technology,

including efficient manufacturing of PMC arrays, will contribute to even shorter processing times, ultimately revolutionizing signal processing applications.

(2) From the perspective of application

Certainly, based on proven silicon-based memory, ADC and neural network computing units, the uncooled thermal imagers coupled with CNN algorithms appear more versatile. The system can be adapted to different application scenarios by using different detectors. But combining the detector with the memory and computation unit results in fast system speeds and eliminates redundant transmission power consumption. These attractive advantages make in-sensor computing worthwhile for further investigation. Until now, Liu et al. worked out the ferroelectric-defined reconfigurable homojunction for in-memory sensing and computing⁸. The response wavelength of the device is within the near-infrared and the response is up to 800 mA/W. Mueller et al. worked out the ultrafast machine vision with WSe₂ neural network image sensors⁹. The response wavelength of the device is within the visible light and the response is up to 50 mA/W. We are the first to combine ultra-fast memory (20 ns) with mid-wave infrared detection (1.68A/W @ 3.5μm). How to achieve the same high responsivity as a single uncooled thermal detector in the process of combining memories is our next goal.

Fig. R2.4 | Speed of Image Inference⁷. a, Accuracy vs time. b, GPU time (milliseconds) for each model.

Revision:

We have added some discussion of photoresponse times in the concluding paragraph. Additionally, corresponding discussions have been incorporated into the revised manuscript as follows: “Based on the 49 μ s rising time of our PMC devices’ photoresponse, the time of image recognition and processing is on the order of microseconds. It is faster than conventional CMOS image processing systems which includes ADC data conversion time (10~50 μ s) and the time of image processing and recognition (~100 ms). In the future, optimizing the response time of the PMC device will make the processing time of the system even shorter.”. (Page 16)

Reference:

1. Safaei, A. et al. Multi-spectral frequency selective mid-infrared microbolometers. *Opt. Express* **26**, 32931-32940 (2018).
2. Pris, A. D. et al. Towards high-speed imaging of infrared photons with bio-inspired nanoarchitectures. *Nature Photonics* **6**, 195-200 (2012).
3. Wei, J. et al. Zero-bias mid-infrared graphene photodetectors with bulk photoresponse and calibration-free polarization detection. *Nature Communications* **11**, 6404 (2020).
4. Zhang, Z., Jiang, J., Chen, C. & Huang, X. The Design of A 12-Bit Two-Step Single Slope ADC For Carbon Based Image Sensors. *2023 IEEE 11th Asia-Pacific Conference on Antennas and Propagation (APCAP)*, 2023: IEEE.
5. Park, H., Yu, C., Kim, H., Roh, Y. & Burm, J. Low Power CMOS Image Sensors Using Two Step Single Slope ADC With Bandwidth-Limited Comparators & Voltage Range Extended Ramp Generator for Battery-Limited Application. *IEEE Sensors Journal* **20**, 2831-2838 (2020).
6. Nie, K., Zha, W., Shi, X., Li, J., Xu, J. & Ma, J. A Single Slope ADC With Row-Wise Noise Reduction Technique for CMOS Image Sensor. *IEEE Transactions on Circuits and Systems I: Regular Papers* **67**, 2873-2882 (2020).
7. Huang, J. et al. Speed/Accuracy Trade-Offs for Modern Convolutional Object Detectors. *Proceedings of the IEEE conference on computer vision and pattern recognition 2017*: IEEE.
8. Wu, G. et al. Ferroelectric-defined reconfigurable homojunctions for in-memory sensing and computing. *Nature Materials* **22**, 1499-1506 (2023).
9. Mennel, L., Symonowicz, J., Wachter, S., Polyushkin, D. K., Molina-Mendoza, A. J. & Mueller, T. Ultrafast machine vision with 2D material neural network image sensors. *Nature* **579**, 62-66 (2020).

Comment 4:

The manuscript highlights an ultrafast (~20 ns) weight update process, reducing weight programming power consumption to ~1.8 fJ. Although emphasized, the discussion lacks depth. Further exploration into how pulse width affects channel conductance and the factors constraining minimum pulse width is necessary.

Response:

We thank the reviewer's comment. The electric field in h-BN and MoS₂ layers is around 14 MV cm⁻¹ when a 30 V or -30 V voltage is applied, which is much larger than the 7.43 MV cm⁻¹ that can induce the FN tunneling effect in the Cr/h-BN/graphene heterostructure. So, the FN tunneling current of carriers should be large in the PMC device and play an important role in the short writing time. Continuing to increase the gate voltage theoretically leads to higher tunnelling currents and program time can be as low as 1 ns¹. However, the minimum pulse width (20 ns) of the instrument (B1500 SPGU) limits further exploration.

We explored further and found that voltage for nanoseconds duration is a high-frequency signal and the parasitic impedances could increase the effective duration of the voltage pulse. In the following context, we used an oscilloscope to measure the actual ultrafast pulses received by the PMC device. The corresponding test results are shown in **Fig. R2.5**.

We have further explored how pulse width affects the regulation of the conductance state. Different pulse settings tend to have a direct effect on the electrical dynamic range of the device, the number of states stored in the device and the average state interval. Here we tested the modulation of some pulses in **Fig. R2.6**. By analyzing the test results, the following main conclusions were summarized in **Table R2.1**. (1) For pulses of the same time width, as the pulse amplitude becomes progressively larger, the conductance modulation range becomes larger and the number of conductance states becomes smaller. (2) For a pulse of the same amplitude, a larger pulse width results in a larger conductance modulation range and a smaller number of conductance states. In the table we quantitatively show the range of conductance states, the number of conductance states and the average conductance interval corresponding to the pulse modulation.

Fig. R2.5 | The voltage pulses after the transmission process. a, c, e, Pulses with 30 V voltage amplitude and pulse width with 10 ns, 20 ns and 50 ns generated by B1500 respectively and captured by oscilloscope. The corresponding effective duration of the pulse (FWHM, full width at half maximum) are 15.9 ns, 21 ns and 48.7 ns, respectively. b, d, f, Pulses with -30 V voltage amplitude and pulse width with 10 ns, 20 ns and 50 ns, generated by B1500, respectively. The corresponding FWHM are 16 ns, 21 ns and 49 ns, respectively.

Fig. R2.6 | Pulse modulation under various pulse set. a-h, Pulse modulation results under pulse set (-15 V, 50 ns), (-15 V, 100 ns), (-17 V, 100 ns), (-18 V, 100 ns), (-18 V, 50 ns), (-19 V, 100 ns), (-19 V, 50 ns) and (-20 V, 100 ns), respectively.

Table R2.1. Pulse modulation results.

V_{GS} and Pulse time	States number	Dynamic range (μA)	average conductance interval (μA)
-15 V, 50 ns	52	0.16	0.003
-15 V, 100 ns	30	0.56	0.0187
-17 V, 100 ns	27	1	0.037
-18 V, 100 ns	15	0.96	0.064
-18 V, 50 ns	22	0.4	0.0182
-19 V, 100 ns	10	1	0.1
-19 V, 50 ns	33	0.7	0.0212
-20 V, 100 ns	8	1.2	0.15

Revision:

We have added **Supplementary Fig. 12** to show the voltage pulses after the transmission process (**Supplementary**, Page 14). Corresponding discussions have been incorporated into the revised manuscript as follows: “*The voltage for nanoseconds duration is a high-frequency signal and the parasitic impedances could increase the effective duration of the voltage pulse. To address this concern, we utilized an oscilloscope to measure the actual ultrafast pulses received by the PMC device. The corresponding test results, illustrating the actual pulse waveform, are shown in **Supplementary Fig. 12**.”. (Page 10) We have added **Supplementary Fig. 13** to show the pulse modulation under various pulse sets and **Supplementary Table 1** to summarise the effects of modulation in three ways. (**Supplementary**, Page 15-17). Corresponding discussions have been incorporated into the revised manuscript as follows: “*Pulse modulation under various pulse sets can be found in **Supplementary Fig. 13**.”. (Page 10)**

Reference:

1. Liu, L. et al. Ultrafast non-volatile flash memory based on van der Waals heterostructures. *Nature Nanotechnology* **16**, 874-881 (2021).

Comment 5:

A lower conductance results in a lower responsivity, which was ascribed to the reduced number of ionizable atoms within the heterojunction channel. This explanation lacks clarity. Defining the number of ionizable atoms and considering interface barriers, especially those of MoS₂-BP and MoS₂-metal, is essential for comprehensively explaining the device's operation. More experimental supports are recommended.

Response:

Thank you for your valuable comment. We did the Kelvin Probe Force Microscope (KPFM) experiment to supplement and improve the explanation of the mechanism. In **Fig. R2.6 a-e**, we show the material interface barriers between (graphene, BN) 160 meV, (BN, MoS₂) 137 meV, (BP, MoS₂) 296 meV, (BP, Au) 47 meV and (MoS₂, Au) 246 meV, respectively. In **Fig. R2.7 f**, we show the relative positions of the Fermi energy levels of the materials before contact with the KPFM test results. This also verifies the correctness of the energy band structure of the manuscript in **Fig. 4**.

The claim of ionizable atoms is not specific and rigorous enough. After further theoretical analysis, “the number ionizable atoms” represents the number of electrons in the covalent electrons between boron and phosphorus atoms in the depletion region of the heterojunction that can be ionized under mid-wave infrared excitation. Firstly, from the photomapping diagram, we can see that the photocurrent is mainly generated by photogenerated electron-hole pairs in the depletion region of the heterojunction under the action of the built-in electric field. When in the low-conductance state, the lower concentration of electrons within MoS₂ leads to a narrower width of the black phosphorus depletion region, so that the number of covalent bonds that can be excited becomes smaller, leading to a weakening of the photocurrent. On the contrary, when in the high conductance state, the higher electron concentration in MoS₂ leads to the widening of the width of the black phosphorus depletion region, so that the number of covalent bonds that can be excited becomes larger, resulting in the enhancement of the photocurrent.

Fig. R2.7 | Kelvin Probe Force Microscope (KPFM) for material interface barriers. a-e, Interface barriers between (graphene, BN) 160 meV, (BN, MoS₂) 137 meV, (BP, MoS₂) 296 meV, (BP, Au) 47 meV and (MoS₂, Au) 246 meV, respectively. f, Alignment of energy bands before contact.

Revision:

We have added **Supplementary Fig. 5** to show the Kelvin Probe Force Microscope (KPFM) for energy band alignment. (**Supplementary**, Page 7). Besides, we have revised the manuscript for more specific theoretical explanations. Corresponding discussions have been incorporated into the revised manuscript as follows: “*Firstly, from the photomapping diagram (see in **Supplementary Fig. 24**), the photocurrent is mainly generated by photogenerated electron-hole pairs in the depletion region of the heterojunction under the action of the built-in electric field. The photogenerated carriers come from the covalent electrons between boron and phosphorus atoms in the depletion region of the heterojunction under mid-wave infrared excitation. In the depicted low-conductance state shown in Fig. 4e, the lower concentration of electrons within MoS₂ leads to a narrower width of the BP depletion region, reducing the number of covalent bonds that can be excited and thereby weakening the photocurrent.*”. “*the higher electron concentration in MoS₂ leads to the widening of the BP depletion region, so that the number of covalent bonds that can be excited becomes larger, resulting in the enhancement of the photocurrent.*” (Page 13-14)

Comment 6:

The device's infrared light response predominantly arises from BP, while the differing dependence of near-infrared and mid-infrared responsivity on the conductance of the device remains unexplained. What is the underlying reason? What impact will this difference have on subsequent computing?

Response:

Thank you for your valuable comment. We have performed several sets of experiments to demonstrate that the dependence of the mid-infrared response on the device conductance is indeed non-linear in **Fig. R2.8**. In the near-infrared region, the photon energy is sufficiently high to directly excite electron-hole pairs, generating a photocurrent. Therefore, the photovoltaic effect predominantly governs this region. This direct photo-generated carrier generation process is linear. In the mid-wave infrared region, the photon energy is lower and more susceptible to thermal effects, often producing photo-thermoelectric and photo-bolometric effects¹⁻³. When photons are absorbed and converted into thermal energy, a local or overall temperature increase occurs in the material, leading to changes in the photocurrent due to temperature variation. This process is influenced by the duration of illumination, introducing non-linear characteristics.

In **Fig. R2.8**, the red curve is a third-order polynomial fit curve. The equation for the fit is $y=0.01762+439619.3x-5.56*10^{11}x^2-295499.9x^3$ ($R^2=0.96936$). Nonlinearity can lead to very complex polynomial fitting relationships. Although it is also able to achieve the programming of the corresponding responsivity through transformation, the process of transformation requires a greater amount of computation, which can lead to an increase in computation time. Besides, areas with larger relative rates of change in responsivity will produce some errors when programming the states.

Fig. R2.8 | The dependence of mid-infrared responsivity on the conductance. The red dots represent the actual blackbody responsivity at different conductance states, and the red curve is a third-order polynomial fit curve.

Revision:

Corresponding discussions have been incorporated into the revised manuscript as follows: “*For the mid-wave infrared, the responsivity does not correspond linearly to the conductance states (Supplementary Fig. 26). Nonlinearity can lead to complex polynomial fitting relationships. Although it is also able to achieve the programming of the corresponding responsivity through transformation, the process of transformation requires a greater amount of computation, which can lead to an increase in computation time.*”. (Page 11)

Reference:

1. Low, T., Engel, M., Steiner, M. & Avouris, P. Origin of photoresponse in black phosphorus phototransistors. *Physical Review B* **90**, (2014).
2. Deng, Y. et al. Black Phosphorus–Monolayer MoS₂ van der Waals Heterojunction p–n Diode. *ACS Nano* **8**, 8292-8299 (2014).
3. Youngblood, N., Chen, C., Koester, S. J. & Li, M. Waveguide-integrated black phosphorus photodetector with high responsivity and low dark current. *Nature Photonics* **9**, 247-252 (2015).

Comment 7:

Material thickness significantly impacts device performance, as noted by the authors. However, the criteria for determining the thickness used in the study are

undisclosed. Comprehensive experimentation exploring the effects of various thicknesses on performance is advisable.

Response:

Thank you for your comments. Through further research on thickness, we found that three typical phenomena existed.

(1) MoS₂ and h-BN are all thicker, with typical values of around 20 nm. This thickness configuration results in a PMC device with a small range of conductance state regulation and no memory window in **Fig. R2.9**.

Fig. R2.9 | The device performance of thicker MoS₂ and h-BN. a, I_{DS} - V_{DS} curve of the PMC device with different V_{BG} . b, I_{DS} - V_{BG} curve of the PMC device with $V_{DS} = 1 V$. c, d, the thickness of MoS₂ (19 nm) and h-BN (20 nm).

(2) Typical values of around 10 nm MoS₂ and 20 nm h-BN. This thickness configuration results in a PMC device with a large range of conductance state regulation (ON/OFF ratio 10^6) and no memory window in **Fig. R2.10**.

Fig. R2.10 | The device performance of MoS₂ (~10 nm) and *h*-BN (~20 nm). a, I_{DS} - V_{DS} curve of the PMC device with different V_{BG} . b, I_{DS} - V_{BG} curve of the PMC device with $V_{DS} = 1V$. c, d, The thickness of MoS₂ (11 nm) and *h*-BN (21 nm).

(3) Typical values of around 20 nm MoS₂ and 10 nm *h*-BN. This thickness configuration results in a PMC device with a small range of conductance state regulation with a considerable memory window in **Fig. R2.11**.

Fig. R2.11 | The device performance of MoS₂ (~20 nm) and h-BN (~10 nm). a, I_{DS} - V_{DS} curve of the PMC device with different V_{BG} . b, I_{DS} - V_{BG} curve of the PMC device with $V_{DS} = 1$ V. c, d, The thickness of MoS₂ (20 nm) and h-BN (13 nm).

Revision:

We have added **Supplementary Fig. 9** to show the effects of various thicknesses on performance (**Supplementary**, Page 11). Corresponding discussions have been incorporated into the revised manuscript as follows: *“the effect of thickness can be seen in **Supplementary Fig. 9**”*. (Page 8)

Comment 8:

What is the mid-infrared detectivity of the device? What is the influence of various conductance on detectivity?

Response:

Thank you for your valuable comment. We rebuilt a batch of devices and tested the noise current (I_{Noise}) and blackbody response in multiple conductance states (36.8

nS to 3.65 μ S) in Fig. R2.12a b, respectively. The specific detectivity (D^*) is defined as $D^* = (S\Delta f)^{1/2} R(I_{noise})^{-1/2}$, where Δf is the electrical bandwidth. Further, we calculated the detectivity of different conductance states in the mid-infrared (3.5 μ m) in Fig. R2.12c. The test results indicate that the maximum specific detectivity is close to 10^9 cm Hz^{1/2} W⁻¹ when the conductance state changes.

Fig. R2.12 | Mid-infrared detectivity of the PMC device. a, Noise current spectral density in multiple conductance states. b, Responsivity as a function of wavelength at various conductance states. Conductance state change from 36.8 nS to 3.65 μ S at $V_{DS} = 0.1$ V. c, Specific detectivity of different conductance states in the mid-infrared (3.5 μ m).

Revision:

We have added **Supplementary Fig. 30** to show the influence of various conductance on detectivity (**Supplementary**, Page 31). Corresponding discussions have been incorporated into the revised manuscript as follows: “*The maximum specific detectivity of the tested device is close to 10^9 cm Hz^{1/2} W⁻¹ in the mid-infrared (3.5 μ m) when the conductance state changes (see **Supplementary Fig. 30**).*” (Page 13)

Comment 9:

Several minor issues are noted:

9.1 Reference No. 26 cited in the manuscript is unrelated to the content. “The photocurrent spectrum of the device in the MWIR band was obtained by Fourier transform infrared spectroscopy (FTIR)26.” “26. Li, T. et al. Developing fibrillated cellulose as a sustainable technological material. 364 Nature 590, 47-56 (2021).”

9.2 Most reference citations come from the authors’ papers. It is recommended

to broaden the reference sources.

9.3 The photocurrent mapping was mentioned in the Methods section. However, the corresponding results can't be found in the manuscript

Response:

Thank you for your valuable comment.

9.1: “26. Li, T. et al. Developing fibrillated cellulose as a sustainable technological material. 364 Nature 590, 47-56” is really unrelated to the content. We apologize for confusion caused to our readers by the incorrect citation of this manuscript. We use “Alves, F. D. P. et al. NIR, MWIR and LWIR quantum well infrared photodetector using interband and intersubband transitions. Infrared Phys. Techn. 50, 182-186 (2007).” to replace it.

9.2: Because the project is an innovation based on the group's work, the manuscript does cite many articles from the same group. Based on the reviewer's suggestion, we deleted some of the reference citations of our own papers. For a more comprehensive study, we broaden the reference sources.

“Martyniuk, P. et al. Infrared avalanche photodiodes from bulk to 2D materials. Light: Science & Applications 12, 212 (2023).” replaces “Wang, Y. et al. Fast uncooled mid-wavelength infrared photodetectors with heterostructures of van der Waals on epitaxial HgCdTe. 34, 2107772 (2022).”

“Liu, L. et al. Ultrafast flash memory with large self-rectifying ratio based on atomically thin MoS₂-channel transistor. Materials Futures 1, 025301 (2022).” replaces “Huang, X. H., Liu, C. S., Tang, Z. W., Zeng, S. F., Wang, S. Y. & Zhou, P. An ultrafast bipolar flash memory for self-activated in-memory computing. Nat. Nanotechnol. 18, 486-492 (2023).”

“Wang, Y. et al. Optoelectronic Synaptic Devices for Neuromorphic Computing. Advanced Intelligent Systems 3, 2000099 (2021).” replaces “Liu, X. X., Wang, S. Y., Di, Z. Y., Wu, H. Q., Liu, C. S. & Zhou, P. An Optoelectronic Synapse Based on Two-Dimensional Violet Phosphorus Heterostructure. Adv. Sci. 10, 9 (2023).”

Add “Park, H., Yu, C., Kim, H., Roh, Y. & Burm, J. Low Power CMOS Image Sensors Using Two Step Single Slope ADC With Bandwidth-Limited Comparators & Voltage Range Extended Ramp Generator for Battery-Limited Application. IEEE Sensors Journal 20, 2831-2838 (2020).”

Add Nie, K., Zha, W., Shi, X., Li, J., Xu, J. & Ma, J. A Single Slope ADC With Row-Wise Noise Reduction Technique for CMOS Image Sensor. *IEEE Transactions on Circuits and Systems I: Regular Papers* 67, 2873-2882 (2020).

Add Zhang, Z., Jiang, J., Chen, C. & Huang, X. The Design of A 12-Bit Two-Step Single Slope ADC For Carbon Based Image Sensors. 2023 IEEE 11th Asia-Pacific Conference on Antennas and Propagation (APCAP); 2023 22-24 Nov. 2023; 2023. p. 1-2.

Add Huang, J. et al. Speed/accuracy trade-offs for modern convolutional object detectors. *Proceedings of the IEEE conference on computer vision and pattern recognition*; 2017; 2017. p. 7310-7311.

Add Gao, T. et al. Temperature-triggered chemical switching growth of in-plane and vertically stacked graphene-boron nitride heterostructures. *Nature Communications* 6, 6835 (2015).

Add Chen, S. et al. Wafer-scale integration of two-dimensional materials in high-density memristive crossbar arrays for artificial neural networks. *Nature Electronics* 3, 638-645 (2020).

Add Liu, G. et al. Graphene-assisted metal transfer printing for wafer-scale integration of metal electrodes and two-dimensional materials. *Nature Electronics* 5, 275-280 (2022).

Add Nakatani, M. et al. Ready-to-transfer two-dimensional materials using tunable adhesive force tapes. *Nature Electronics* 7, 119-130 (2024).

9.3: The photocurrent mapping was deleted by mistake. We have included photocurrent mapping at 1550 nm laser in **Fig. R2.13**. This result indicates that the device photocurrent mainly originates from the heterojunction region of BP/MoS₂.

Fig. R2.13 | Photocurrent mapping at 1550 nm laser. a, Image of the device under a 50x optical microscope, scale 5 μm. The red dotted box represents the area used for photocurrent scanning. b, Photocurrent mapping at 1550 nm. The solid blue box represents the location of the MoS₂. The solid red box represents the location of BP. The blue dashed box represents the position of the metal electrode. The right panel is the net photocurrent intensity spectrum.

Responses to Reviewer # 3:

The manuscript describes a novel Van der Waals (VdWs) photodetector for Mid-wave infrared (MWIR) detection, which fully demonstrates the unique properties of VdWs materials. The work is very interesting and will further promote the application of VdWs materials for MWIR detection and recognition with heterojunction and high performance. Among the previously proposed works, devices used for multiply-accumulate operations require additional memory to store weights and do not allow for efficient functional integration. Zhu et al in this paper realized the in-sensor computing hardware that integrates MWIR photodetector with adjustable responsivity, memory, and computing simultaneously. The device has the potential to realize recognition of objects at night through MWIR detection with lower programming energy consumption. The results are convincing in the sense that they provide sufficient evidence that MWIR photodetector has non-volatile memory characteristics including dynamic range, maintenance, and durability. The authors also measured the linear correspondence between different conductance and responsivity in real-time. The measurement of blackbody radiation proved the possibility of practical use of the device. Hence, I recommend the paper for publication in Nature Communications if the authors will address the following questions:

Response:

Thank the reviewer for carefully reading the manuscript and positive comments on our work ‘*I recommend the paper for publication in Nature Communications*’. Meanwhile, the reviewer has concerns about pulse modulation, stability of responsivity and device structure. Based on these critical and constructive comments, the new experimental data have been added to the revised manuscript to enhance the readability. With the help of the reviewer, the whole manuscript has been largely improved. In the following, we will address all comments point-by-point and revised the manuscript. We hope that the revised manuscript would remove the reviewers’ concerns.

Comments 1:

The ultrafast 20 ns pulse generated by the B1500 is usually not an ideal square wave. The authors should provide the actual pulse applied on the device through the oscilloscope.

Response:

Thank you for your valuable comment. Incorporating the reviewers' suggestions, we explored further and found that voltage for nanoseconds duration is a high-frequency signal and the parasitic impedances could increase the effective duration of the voltage pulse. To address this concern, we utilized an oscilloscope to measure the actual ultrafast pulses received by the PMC device. The corresponding test results, illustrating the actual pulse waveform, are shown in **Fig. R3.1**. As described, the B1500 was utilized to generate pulses with a voltage amplitude of 30 V and pulse widths of 10 ns, 20 ns, and 50 ns, respectively. Subsequently, an oscilloscope was employed to capture the actual pulse curves of the outputs. Upon analysis, the corresponding effective pulse durations (FWHM, full width at half maximum) were determined to be 15.9 ns, 21 ns, and 48.7 ns, respectively. These findings indicate that achieving a 10 ns pulse output at high voltage (30 V) is challenging, whereas 20 ns and 50 ns pulses can be achieved more readily. We believe these measurements provide a clearer understanding of the pulse characteristics and enhance the reliability of our experimental setup.

Fig. R3.1 | The voltage pulses after the transmission process. a, c, e, Pulses with 30 V voltage amplitude and pulse width with 10 ns, 20 ns and 50 ns generated by B1500 respectively and captured by oscilloscope. The corresponding effective duration of the pulse (FWHM, full width at half maximum) are 15.9 ns, 21 ns and 48.7 ns, respectively. b, d, f, Pulses with -30 V voltage amplitude and pulse width with 10 ns, 20 ns and 50 ns, generated by B1500, respectively. The corresponding FWHM are 16 ns, 21 ns and 49 ns, respectively.

Revision:

We have added **Supplementary Fig. 12** to show the voltage pulses after the transmission process (**Supplementary**, Page 14). Corresponding discussions have been incorporated into the revised manuscript as follows: *“The voltage for nanoseconds duration is a high-frequency signal and the parasitic impedances could increase the effective duration of the voltage pulse. To address this concern, we utilized an oscilloscope to measure the actual ultrafast pulses received by the PMC device. The corresponding test results, illustrating the actual pulse waveform, are shown in **Supplementary Fig. 12.**”*. (Page 10)

Comments 2:

The amplitude and pulse width of the pulses are expected to have varying effects on the memory state. To comprehensively understand the pulse modulation effectiveness, it is recommended that the authors provide a range of pulse modulations with different amplitudes and pulse widths.

Response:

Thank the reviewer for the valuable comment. As the reviewer says, “the amplitude and pulse width of the pulses are expected to have varying effects on the memory state”. Different pulse settings tend to have a direct effect on the electrical dynamic range of the device, the number of states stored in the device and the average state interval. Here we tested the modulation of some pulses in **Fig. R3.2**. By analyzing the test results, the following main conclusions were summarized in **Table R3.1**. (1) For pulses of the same time width, increasing pulse amplitude results in a larger conductance modulation range and a smaller number of conductance states. (2) For pulses of the same amplitude, increasing pulse width leads to a larger conductance modulation range and a smaller number of conductance states. In the table, we quantitatively show the range of conductance states, the number of conductance states and the average conductance interval corresponding to the pulse modulation. These findings contribute to a comprehensive understanding of the impact of pulse parameters on device performance.

Fig. R3.2 | Pulse modulation under various pulse sets. a-h, Pulse modulation results under pulse sets (-15 V, 50 ns), (-15 V, 100 ns), (-17 V, 100 ns), (-18 V, 100 ns), (-18 V, 50 ns), (-19 V, 100 ns), (-19 V, 50 ns) and (-20 V, 100 ns), respectively.

Table R3.1. Pulse modulation results.

V_{GS} and Pulse time	States number	Dynamic range (μA)	average conductance interval (μA)
-15 V, 50 ns	52	0.16	0.003
-15 V, 100 ns	30	0.56	0.0187
-17 V, 100 ns	27	1	0.037
-18 V, 100 ns	15	0.96	0.064
-18 V, 50 ns	22	0.4	0.0182
-19 V, 100 ns	10	1	0.1
-19 V, 50 ns	33	0.7	0.0212
-20 V, 100 ns	8	1.2	0.15

Revision:

We have added **Supplementary Fig. 13** to show the pulse modulation under various pulse sets and **Supplementary Table 1** to summarise the effects of modulation in three ways. (**Supplementary**, Page 15-17). Corresponding discussions have been incorporated into the revised manuscript as follows: “*Pulse modulation under various pulse sets can be found in **Supplementary Fig. 13**.*”. (Page 10)

Comments 3:

Why the device structure should be a semi-floating gate and what will happen if it is stacked directly vertically. The authors need to provide test results for direct stacking.

Response:

Thank the reviewer for the valuable comment. After experiments, we found that the device structure of semi-floating gate can effectively regulate the threshold of the transistor to achieve nice memory performance. But direct stacking leads to an overlap between the built-in electric field of the heterojunction and the electric field generated by the floating gate storage charge. This results in a weaker threshold regulation of the transistor by the floating gate charge. The different memory characteristics of the two devices are demonstrated in the **Fig. R3.3**.

Fig. R3.3 | Memory performance comparison between semi-floating gate structure and directly stacked structure. a, Device structure diagram of semi-floating gate. b, Diagram of directly stacked device structure. c, Device memory performance of semi-floating gate. b, Device memory performance of directly stacked device structure.

Revision:

We have added **Supplementary Fig. 10** to show the memory performance comparison between semi-floating gate structure and directly stacked structure. (**Supplementary, Page 12**).

Comments 4:

The position of photocurrent generation is very helpful in exploring the mechanism of photocurrent, and the authors should provide photocurrent mapping images at a specific wavelength.

Response:

We thank the reviewer for the valuable comment. In response to the suggestion,

we have included photocurrent mapping at 1550 nm laser in **Fig. R3.4**. This result indicates that the device photocurrent mainly originates from the heterojunction region of BP/MoS₂.

Fig. R3.4 | Photocurrent mapping at 1550 nm laser. a, Image of the device under a 50x optical microscope, scale 5 μm. The red dotted box represents the area used for photocurrent scanning. b, Photocurrent mapping at 1550 nm. The solid blue box represents the location of the MoS₂. The solid red box represents the location of BP. The blue dashed box represents the position of the metal electrode. The right panel is the net photocurrent intensity spectrum.

Revision:

Corresponding discussions have been incorporated into the revised manuscript as follows: “see *Supplementary Fig. 24*.”. (Page 21)

Comments 5:

It has been demonstrated in the paper that the device has a stable responsivity under irradiation with different power densities of 1550 nm laser. The authors need to provide test results at 1310 nm to demonstrate the consistency of the conclusions.

Response:

Thank the reviewer for the valuable comment. In the original manuscript, we conducted tests on the dependence of the net photocurrent on the laser power for different conductance states at 1550 nm wavelength (**Fig. 3d**) and determined the power range for constant responsivity. Additionally, the inset of **Fig. 3e** illustrates the stable

responsivity in different states. To further demonstrate the reliability as well as the consistency of the conclusions, we tested the experimental results at a wavelength of 1310nm in **Fig. R3.6** and we also retested the experimental results at a wavelength of 1550 nm in **Fig. R3.5**. These additional figures provide comprehensive evidence supporting the stability and consistency of our findings across different wavelengths.

Fig. R3.5 | Stable responsivity of different conductance states at 1310 nm laser. The dots are the actual measurements and the dashed line is the linear fit curve.

Fig. R3.6 | Stable responsivity of different conductance states at 1550 nm laser. The dots are the actual measurements and the dashed line is the linear fit curve.

Revision:

We have added **Supplementary Fig. 23** to show the stable responsivity of different conductance states at a 1310 nm laser. (**Supplementary**, Page 26). And we replaced **Fig. 3d** with **Fig. R3.5**. (Page 27)

Comments 6:

In Figure 5b, a small difference occurs in the responsivity replacement software weights of the devices. The author should add the weight values in tabular form to make it clearer.

Response:

We appreciate the valuable comment from the reviewer. In order to enhance the clarity of the data, we have included the weighting information in tabular form as suggested. The tables, designated as **Table R3.2** and **Table R3.3**, provide a clearer presentation of the data.

Table R3.2 Trained weight.

0.02831	0.01813	0.0159	0.00289	-0.05268	-0.04726	0.01888
0.00816	0.02476	0.07657	0.09053	-0.01876	-0.07937	-0.01499
-0.02725	-0.06584	0.01985	0.18494	0.11569	-0.08993	-0.09427
-0.00957	-0.07922	-0.12999	0.09536	0.27019	0.09944	-0.0146
0.02329	-0.04632	-0.1831	-0.15598	0.05369	0.08355	0.03727
0.04218	0.04595	-0.04669	-0.12698	-0.02087	0.03328	0.02522
0.01452	0.0351	0.00392	-0.06041	-0.01689	0.01182	0.00491

Table R3.3 PMC device weight.

0.02729	0.02056	0.01384	0	-0.05216	-0.0488	0.02056
0	0.02729	0.07704	0.09053	-0.02056	-0.07704	-0.01384
-0.02729	-0.06628	0.02056	0.1678	0.10124	-0.10124	-0.10124
0	-0.07704	-0.10124	0.10124	0.27	0.10124	-0.01384
0.02056	-0.0488	-0.1678	-0.1678	0.05216	0.07704	0.03804
0.03804	0.0488	-0.0488	-0.10124	-0.02056	0.03468	0.02729
0.01384	0.03468	0	-0.06628	-0.02056	0.00981	0

Revision:

We have added **Supplementary Table 3, 4** to show specific weighting values (**Supplementary**, Page 41). Corresponding discussions have been incorporated into the revised manuscript as follows: “*Weight values are also listed in tabular form in **Supplementary Table 3, 4.***”. (Page 29)

Based on the reviewer's comments, we have performed the following correction:

Revised manuscript

- 1, Page1, we change the title to “Non-volatile photodiodes in the near- to mid-infrared region”.
- 2, Page2, we removed uncritical descriptions of the simulation.
- 3, Page8, we added study of device material thickness.
- 4, Page9 and 10, we added description of the actual pulse profile and retention mechanism.
- 5, Page11 and 12, we re-tested the responsivity stability and the non-linear dependence of the mid-wave infrared responsivity on the conductance state.
- 6, Page13 and 14, we tested the detectivity of the devices at different conductance states and summarised the dependence of the detectivity on the conductance state. We also tested the KPFM to further detail the intrinsic mechanism of the device.
- 7, Page15, we quantified the proportion of substitution. We extended the range of convolutional layer substitutions and analyzed the proportion with 0%, 33.3%, 66.7%, and 100% substitution. The PMC hardware responsivity weights can reach 89% mean Average Precision index of the feature extraction network software weights.
- 8, Page16, The speed advantages of the devices and the prospects for large-scale preparation are described in our summary.

Responses to Reviewers' Comments

We acknowledge the reviewers for carefully reading our manuscript '*Non-volatile photodiodes in the near- to mid-infrared region* (NCOMMS-24-12907A)' and providing constructive comments on our work. According to the reviewers' comments, we have carefully revised our manuscript and provided more detailed data to improve the manuscript's readability. We reorganized the manuscript and Supplementary Information to present more detailed description of the concerns. With the help of the reviewers, we believe that the revision has been significantly improved. The corresponding revisions concerning comments have been provided and highlighted in red in the revised manuscript and Supplementary Information. The corresponding responses of the reviewers are marked by blue words. The detailed revisions in the revised manuscript are listed on a separate page at the end of the response letter.

Responses to Reviewer # 1:

General Comments:

I have carefully reviewed the authors' responses and am pleased to note that most of my previous questions have been comprehensively addressed. The provided information is clear, detailed, and meets the expected standards. However, I still have a significant concern regarding the use of the reported PMC devices as the black radiation detectors in the memory CNN described by the authors. For a device to be effective in this context, it is critical that its responsivity remains linearly proportional to the stored conductance. Deviations from this linearity can lead to increased computation time and reduced efficiency. In the MWIR or black radiation task presented in the revised manuscript, the device exhibits a nonlinear behavior, which is nonideal for CNN applications within the MIR range. This nonlinearity undermines the motivation for using the PMC device for MWIR object detection tasks, as it can compromise performance and computational efficiency. I believe the authors should address this issue and provide a resolution for the nonlinearity of the PMC device's responsivity before the manuscript can be considered for publication in Nature Communications.

Response:

Thank the reviewer for carefully reading the manuscript and positive comments on our work ‘The provided information is clear, detailed, and meets the expected standards.’. Meanwhile, the reviewer has concerns about the non-linear dependence of the mid-wave infrared on the conductance state. We hope that the revised manuscript would remove the reviewers’ concerns.

Thank you for your valuable comment. Theoretically, the photon energy is sufficiently high in the near-infrared region to directly excite electron-hole pairs, generating a photocurrent. Therefore, the photovoltaic effect predominantly governs this region. This direct photo-generated carrier generation process is linear. In the mid-wave infrared region, the photon energy is lower and more susceptible to thermal effects, often producing photo-thermoelectric and photo-bolometric effects¹⁻³. When photons are absorbed and converted into thermal energy, a local or overall temperature increase occurs in the material, leading to changes in the photocurrent due to temperature variation. This process is influenced by the duration of illumination, introducing non-linear characteristics. From this perspective, rapid cooling facilities outside the system can effectively reduce non-linearities.

From the application of neural networks, most of the neural network hardware replacements are ex-situ training⁴⁻⁶, where the trained weights are used to program the device one-time for the corresponding neural network scenario. In fact, it is the number of states that determines accuracy. Even if the correspondence is non-linear, theoretically if the device weights are analogue and all values of the dynamic range are available, then there is no error in replacing the software trained weights. For example, Mennel, L. et al achieved a continuously adjustable responsivity at the expense of additional memory units, more gate voltage energy consumption, and the accuracy is theoretically infinite bits even if there is a non-linear correspondence between responsivity and gate voltage. Both supervised and unsupervised learning and training the sensor classify and encode images with a throughput of 20 million bins per second⁷. The advantages of the PMC system remain significant.

A linear correspondence is the ideal result to pursue for in-situ training. In-situ training requires constant comparison between the device and the theoretical output values, using feedback algorithms to iterate the weights of the device. The correspondence is linear, then a simple linear mapping will be formed, non-linear only increases the amount of computation of the outside trainer during the training process,

but the efficiency of the neural network application of the PMC system will not be reduced because the PMC system is used in the process of detection and recognition with already trained weights, which are one-time programming without the need for multiple iterations of the trainer. The application simulation in the paper is also the result of using the weight substitution of the mid-infrared nonlinearities, and the substitution result is satisfied. The pursuit of linear correspondence is to lay the foundation for the future comprehensive replacement of the training process to achieve in-situ training by using the integrated device of memory and computing.

The solution is to get the mapping function by using a fitting method^{8, 9}. In **Fig. R1.1**, The red dots are the data from the test, and the red line is an exponential fit to the test data. $R^2 = 0.99013, 0.94944, 0.98469$ and 0.98597 , respectively.

Fig. R1.1 | The photoresponse of the PMC device3 at various conductance states under mid-wave infrared laser stimulation. a, b, c, d, The responsivity increases with the increasing conductance state under 2611 nm, 3098 nm, 3403 nm, and 3622 nm, respectively. The red dots are the data from the test, and the red line is an exponential fit ($y = A \exp(-x/t) + B$) to the test data. $R^2 = 0.99013, 0.94944, 0.98469$ and 0.98597 , respectively.

Revision:

We have reorganized **Supplementary Fig. 26** the resolution for the nonlinearity of the PMC device's responsivity (Supplementary, Page 28). Corresponding discussions have been incorporated into the revised manuscript as follows: “*For the mid-wave infrared range, the responsivity does not correspond linearly to the conductance states. This non-linearity increases the computational effort of in-situ training, thereby extending the time to train the neural network. An exponential fitting approximation can be used to establish a fixed function mapping relationship, which will facilitate the mapping of weights and improve training efficiency (see Supplementary Fig. 26)*”. (Page 11)

Reference:

1. Deng, Y. et al. Black Phosphorus–Monolayer MoS₂ van der Waals Heterojunction p–n Diode. *ACS Nano* **8**, 8292-8299 (2014).
2. Low, T., Engel, M., Steiner, M. & Avouris, P. Origin of photoresponse in black phosphorus phototransistors. *Physical Review B* **90**, (2014).
3. Youngblood, N., Chen, C., Koester, S. J. & Li, M. Waveguide-integrated black phosphorus photodetector with high responsivity and low dark current. *Nature Photonics* **9**, 247-252 (2015).
4. Alibart, F., Zamanidoost, E. & Strukov, D. B. J. N. c. Pattern classification by memristive crossbar circuits using ex situ and in situ training. **4**, 2072 (2013).
5. Yakopcic, C., Hasan, R. & Taha, T. M. Memristor based neuromorphic circuit for ex-situ training of multi-layer neural network algorithms. 2015 International Joint Conference on Neural Networks (IJCNN); 2015: IEEE; 2015. p. 1-7.
6. Hasan, R., Yakopcic, C., Taha, T. M. J. A. I. C. & Processing, S. Ex-situ training of large memristor crossbars for neural network applications. **99**, 1-10 (2019).
7. Mennel, L., Symonowicz, J., Wachter, S., Polyushkin, D. K., Molina-Mendoza, A. J. & Mueller, T. Ultrafast machine vision with 2D material neural network image sensors. *Nature* **579**, 62-66 (2020).
8. Chen, P. Y., Peng, X. & Yu, S. NeuroSim+: An integrated device-to-algorithm framework for benchmarking synaptic devices and array architectures. 2017 IEEE International Electron Devices Meeting (IEDM); 2017 2-6 Dec. 2017; 2017. p. 6.1.1-6.1.4.
9. Wu, G. et al. Ferroelectric-defined reconfigurable homojunctions for in-memory sensing and computing. *Nature Materials* **22**, 1499-1506 (2023).

Responses to Reviewer # 2:

General Comments:

The authors have addressed several of my concerns; however, two issues remain unclear.

Response:

Thank the reviewer for carefully reading the manuscript and positive comments on our work ‘The authors have addressed several of my concerns’. We are pleased that under the guidance of the reviewer, the manuscript is more clearly presented on several issues. Meanwhile, the reviewer has concerns about the advantages compared to conventional imaging systems and some details. We hope that the revised manuscript would remove the reviewers’ concerns.

Comment 1:

The authors stated that the overall time of conventional image processing systems exceeds 100 ms. This includes the photoresponse time of an uncooled mid-wave imager (100 μ s to 25 ms), ADC data conversion time (approximately 10-50 μ s), and the reasoning time for image processing and recognition using computer software (at least 100 ms). A key point is that the photoresponse time of commercial uncooled MCT MIR detectors can reach the order of nanoseconds, which is significantly faster than the presented PMC devices (49 μ s). Another key point is that image processing and recognition with the presented PMC devices still require the assistance of computer software, necessitating a reasoning time of at least 100 ms. Therefore, the PMC device does not exhibit any speed advantage. It is suggested to provide a clearer comparison between the PMC device and conventional image processing systems.

Response:

Thank the reviewer for the valuable comment. As the reviewer said the response speed (μ s) of the PMC device is not outstanding compared to commercial uncooled MCT MIR detectors (ns). The image processing systems using PMC devices (Core requirement: non-volatile and tunable responsivity, which traditional detectors do not

have) have speed advantages compared to conventional image processing systems. We will provide a clearer comparison between PMC image processing systems and conventional image processing systems in combination with **Fig. R2.1**.

The conventional image processing system is displayed in **Fig. R2.1a**. Memory and computer architectures are based on the traditional Von Neumann architecture. The whole image processing process includes five processes: (1) Image sensing. The speed depends mainly on the response time of the device. Energy consumption depends mainly on supply voltage and operating current. (2) ADC converts the analogue signals generated by the sensor into digital signals that can be processed by the computer. (3) Memory writes the image data. Performance is largely dependent on memory bandwidth and power consumption. (4) The processor loads the image data (Read speeds depend on memory bandwidth due to the drawbacks of architectural memory walls) and (5) the neural network processes the image data. The process depends mainly on the type of neural network and the performance of the processor.

PMC image recognition system is displayed in **Fig. R2.1b**. This system integrates memory (non-volatile devices) and image processing algorithms (topology of circuit connections) into the sensor array. The device array performs matrix-vector multiplication (MVM) and multiply-and-accumulation (MAC) operations based on $I = RP$ and Kirchhoff's law while sensing light. The operations of the processor reasoning process are arithmetically equivalent to the computation of the device array when sensing the image. The result of the recognition is embedded in the relative magnitude of the output currents in each column (The ability to differentiate between current magnitudes is due to the programming of device responsivity). Each column of current corresponds to an object, when the current in column x is the maximum value and the rest of the columns are almost zero, then the object corresponding to the current in column x is the recognition result (current-driven LED can show the recognition result.). Thus, the reasoning process does not require the assistance of computer software. At the same time, it eliminates time and energy consumption of the ADC, data transmission between memory and processor, and ADC and memory. The neural network simulation in the manuscript is a simulation using the device weights to verify the feasibility of the device applied to the PMC image processing system (The main requirements are the number of programmable states and dynamic range of the device). Mennel, L. et al achieved ultrafast machine vision with 2D material neural network image sensors using this processor architecture. They demonstrated correct pattern classification within ~50

ns¹. In the RedEye design, the convolutional operations are directly implemented in the analogue domain through a charge-sharing tunable capacitor. The energy consumption per frame is 44.3% and 45.6% lower than that for a GPU and CPU, respectively².

Finally, we provide a comparison between the PMC image processing system and the commercial uncooled MCT MIR conventional image processing system in **Table R2.1**.

Fig. R2.1 | Comparison of structure schematics between PMC system and conventional image processing systems³. a, Conventional image recognition system. b, PMC image recognition system.

Table R2.1 Comparison between the PMC image processing system and commercial uncooled MCT MIR conventional image processing system

	Sensor					ADC conversion	Memory write/frame	Processor load image / frame	Process of reasoning	Programming time / energy	System time / energy
	Response	Device time	Effective area/pixel	Cost	Substrate						
PMC image processing system	1.68A/W	49 μ s	\sim 100 μ m ²	Cheap	SiO ₂ /Si	\	\	\	Within the sensing process	\sim 200 μ s / \sim 180nJ	>249 μ s / >180nJ
Conventional	\sim 1A/W	120 ns	1 mm ²	Expensive ¥ 12503 / pixel	CdZnTe	10-50 μ s / \sim 60mW	0.2 ms / 3.3W	10 μ s / \sim 5W	100 ms / \sim 6W	\	>100 ms / >600mJ

* Take the processing of a 1MB image as an example.

* The data of the commercial uncooled MCT detector used in conventional image processing system comes from VL5T0 (THORLABS).

* The memory in conventional image sensors uses data from Solid-State Disk (SSD).

Revision:

We have added Supplementary Table 5 to show the comparison between the PMC image processing system and the commercial uncooled MCT MIR conventional image processing system (Supplementary, Page 41). Additionally, corresponding discussions have been incorporated into the revised manuscript as follows: “*details compared to conventional image processing system with commercial uncooled MCT MIR detectors are displayed in **Supplementary Table 5***”. (Page 16)

Reference:

1. Mennel, L., Symonowicz, J., Wachter, S., Polyushkin, D. K., Molina-Mendoza, A. J. & Mueller, T. Ultrafast machine vision with 2D material neural network image sensors. *Nature* **579**, 62-66 (2020).
2. LiKamWa, R., Hou, Y., Gao, J., Polansky, M. & Zhong, L. J. A. S. C. A. N. Redeye: analog convnet image sensor architecture for continuous mobile vision. **44**, 255-266 (2016).
3. Zhou, F. & Chai, Y. J. N. E. Near-sensor and in-sensor computing. **3**, 664-671 (2020).

Comment 2:

The responsivity of the PMC device at 3600 nm is 1.68 A/W (Fig. 4b), while the maximum responsivity at 3622 nm is lower than 0.28 A/W (Supplementary Fig. 26d). How should this difference be understood?

Response:

Thank the reviewer for the valuable comment. We reanalyzed the data and found that the mid-wave laser was at a high power (156 mW). High laser power during testing can lead to a decrease in device responsivity, which is mainly due to the recombination kinetics of photocarriers involving interactions between trap states and photogenerated carrier.^{1,2} Another reason is that this batch of devices is not the same batch of devices as those in the manuscript, and there are differences between the devices due to different material qualities and production processes. The device in the manuscript is the better performing device. For photoelectric performance there are mainly the following devices: (1) PMC in the manuscript (2) PMC2 for repeated validation of responsivity stability at different wavelengths for different conductance states in **Supplementary Fig. 23**. (3) PMC3 for validation of the photoresponse of a mid-wave laser in relation to the conductance state in **Supplementary Fig. 26**. (Because of the nonlinearity in the correspondence of the blackbody response in the manuscript, we made another batch of device to verify that the mid-wave laser is consistent with the blackbody.) (4) PMC4 for correspondence between detectivity and conductance states in **Supplementary Fig. 30**. We will label the device serial number in Supplementary Information.

Revision:

Additional information is added: “PMC device2 for repeated validation of responsivity stability at different wavelengths for different conductance states at 1310 nm laser.” (Supplementary, Page 26); “The photoresponse of the PMC device3 at various conductance states under mid-wave infrared laser stimulation.” (Supplementary, Page 29); “We rebuilt a batch of devices (PMC4) and tested the noise and blackbody response in multiple conductance states” (Supplementary, Page 32)

Reference:

1. Lopez-Sanchez, O., Lembke, D., Kayci, M., Radenovic, A. & Kis, A. Ultrasensitive photodetectors based on monolayer MoS₂. *Nature Nanotechnology* **8**, 497-501 (2013).
2. Buscema, M., Groenendijk, D. J., Blanter, S. I., Steele, G. A., van der Zant, H. S. J. & Castellanos-Gomez, A. Fast and Broadband Photoresponse of Few-Layer Black Phosphorus Field-Effect Transistors. *Nano Letters* **14**, 3347-3352 (2014).

Responses to Reviewer # 3:**General Comments:**

The authors have addressed all the concerns and this reviewer would like to recommend this revised manuscript for publication in Nature Communications.

Response: We thank the reviewer for recommending the publication of our work in Nature Communications.